# A Benchmark for Deep Information Synthesis

**Debjit Paul**[1]**, Daniel Murphy**[2]**, Milan Gritta**[1]**, Ronald Cardenas**[1]**,**
**Victor Prokhorov**[1]**, Jun Wang**[3]**, Gerasimos Lampouras**[1]

Dataset Contributors:

**Lena Sophia Bolliger**[4]**, Aysim Toker**[1]**, Roy Miles**[1]**, Andreea-Maria Oncescu**[1]**, Jasivan**
**Alex Sivakumar**[5]**, Philipp Borchert**[1]**, Ismail Elezi**[1]**, Meiru Zhang**[6]**, Ka Yiu Lee**[1]**, Guchun Zhang**[1]

[1]Huawei Noah's Ark Lab, UK    [2]Imperial College London    [3]UCL Centre for Artificial Intelligence
[4]University of Zurich    [5]University of Sheffield    [6]University of Cambridge

## Abstract

Large language model (LLM)-based agents are increasingly used to solve complex tasks involving tool use, such as web browsing, code execution, and data analysis. However, current evaluation benchmarks do not adequately assess their ability to solve real-world tasks that require synthesizing information from multiple sources and inferring insights beyond simple fact retrieval. To address this, we introduce DeepSynth, a novel benchmark designed to evaluate agents on realistic, time-consuming problems that combine information gathering, synthesis, and structured reasoning to produce insights. DeepSynth contains 120 tasks collected across 7 domains and data sources covering 67 countries. DeepSynth is constructed using a multi-stage data collection pipeline that requires annotators to collect official data sources, create hypotheses, perform manual analysis, and design tasks with verifiable answers. When evaluated on DeepSynth, 11 state-of-the-art LLMs and deep research agents achieve a maximum F1 score of 8.97 and 17.5 on the LLM-judge metric, underscoring the difficulty of the benchmark. Our analysis reveals that current agents struggle with hallucinations and reasoning over large information spaces, highlighting DeepSynth as a crucial benchmark for guiding future research.

## 1 Introduction

*Information synthesis* involves collecting information from multiple sources and reasoning over it to form coherent insights. While this capability has been central to human decision-making and has driven advances in fields ranging from scientific discovery to policy development (Tricco et al., 2011; Sambre & Brône, 2002), it has traditionally been laborious and cognitively demanding. For example, a travel agency from Singapore might want to know *"Which non-ASEAN countries experienced a significant post-COVID recovery — reaching at least 95% of their 2019 visitor arrival levels to Singapore by 2023, and what were the main reasons for travel (business or tourism)?"* a question that requires identifying ASEAN countries, extracting multiple arrival data from various sources, and analysing them to determine the answer (see Figure 1). Recent Large Language Model (LLM)-based agents with capabilities for reasoning, tool use, and interaction across diverse environments have shown promise in complex tasks, such as for hard-to-find information (Wei et al., 2025), interacting with websites (Lu et al., 2025), and planning to navigate the web (Abuelsaad et al., 2024). However, these developments mostly improve the information-seeking capabilities of agents. It remains crucial to evaluate whether such agents can solve real-world tasks that require synthesizing information from multiple sources and inferring insights beyond simple fact retrieval.

Despite the substantial promise of LLM-based agents for addressing real-world tasks, most existing benchmarks primarily emphasise shallow fact retrieval tasks (Wei et al., 2024), artificial information-seeking questions (Wei et al., 2025; Mialon et al., 2023), or tasks that require information from a single, particularly well-known source like Wikipedia (Wolfson et al., 2025). Furthermore, most agentic benchmarks focus on English-language sources (Wei et al., 2025; Mialon et al., 2023) and

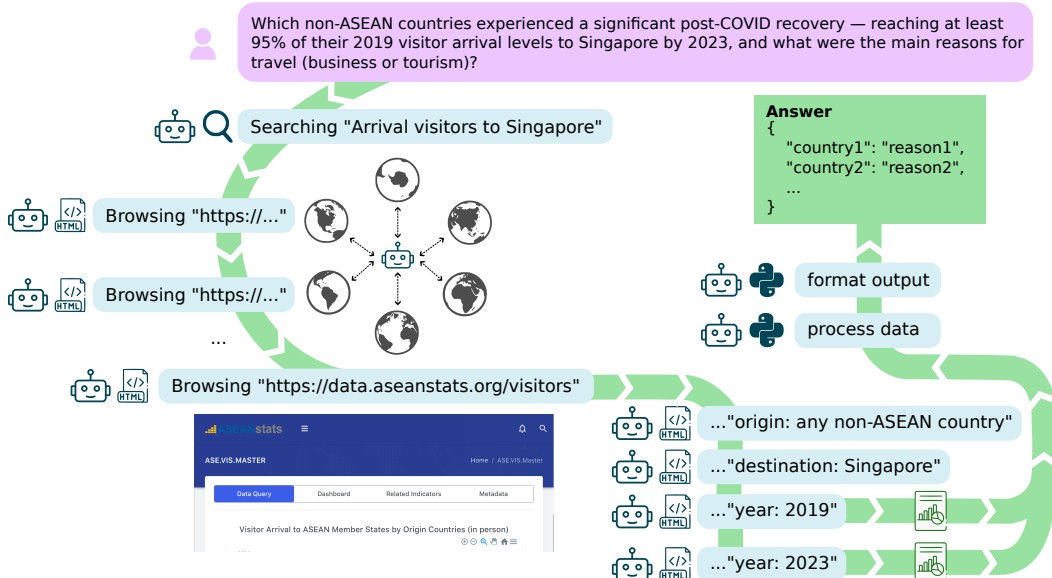

Figure 1: A sample task from DEEPSYNTH, illustrating that synthesizing knowledge requires agents to perform multiple steps, including web browsing, gathering information from multiple sources, reasoning over them, and generating the final answer.

overlook the diversity of regional contexts, languages, and information ecosystems, limiting their ability to evaluate agent performance in realistic, globally distributed settings.

To resolve this gap, we introduce DEEPSYNTH benchmark, a new benchmark comprising 120 challenging and diverse tasks, aimed to evaluate the ability of agents to browse the entire web, combine information from unstructured and structured sources (paragraphs and tables) across 67 countries, and perform analysis to synthesize new information and insights. DEEPSYNTH tasks are annotated with a gold standard, manually annotated reasoning chain, which includes all the intermediate steps, answers and all required supporting evidence. Each task requires agents to navigate an average of 4.2 web pages, and read between 1 to 15 documents and/or tables. These tasks are designed to reflect real-world analysis and insight generation, with an emphasis on the time-intensive nature of processing and integrating information (see Figure 1).

To construct DEEPSYNTH, we asked 16 experts to first curate relevant topics and data sources targeting various countries (§2.2), and subsequently formulate possible hypotheses for each of these data sources. Based on the hypotheses, the experts conducted analyses and derived insights. Finally, drawing on their analyses, they formulated the corresponding questions, answers, and step-by-step reasoning chains. In our experiments, we find that state-of-the-art LLMs struggle significantly on DEEPSYNTH, including recent AI reasoning models such as GPT-5.2-Pro and DeepSeek-R1 (OpenAI, 2025b; Guo et al., 2025). The best-performing model, GPT-5.2-Pro, achieves only an F1 score of 8.70, and 6.25 on the stricter EM metric. We also analyse the performance of specialised deep research agents, i.e. o3-deep-research(OpenAI, 2025a), smolagents(Roucher et al., 2025), and OWL (Hu et al., 2025), and observe they successfully solve only three out of 120 tasks, further underscoring the difficulty of our benchmark. Our analysis reveals that (i) these agents frequently commit navigation and synthesis errors, and (ii) their performance drops sharply when tasks require synthesising information from under-represented sources, e.g. data pertaining to the African region.

To summarize, our main contributions are:

1. We release DEEPSYNTH, a new benchmark for agents that contains 120 real-world and time-consuming information synthesis tasks. [1].

---

[1] Our data and code for DeepSythn Bench are publicly available

2. We show that DEEPSYNTH poses a significant challenge for state-of-the-art agents, revealing key limitations in their capabilities. The best-performing agent achieves only an F1 score of 8.97 points, leaving substantial room for improvement.

3. We conduct an in-depth analysis to explain how DEEPSYNTH is challenging and demonstrate why current agents cannot yet be considered reliable systems for information synthesis.

## 2 THE DEEPSYNTH BENCHMARK

DEEPSYNTH is a benchmark designed to evaluate agents on realistic, time-consuming tasks that require planning, information gathering, and synthesis from the web. Specifically, DEEPSYNTH evaluates agents on their ability to navigate multiple websites, extract information from both structured and unstructured sources, and reason effectively to produce correct solutions. It consists of 120 tasks that are carefully designed and annotated by experts. Each task (see Figure 1) is formulated to yield a concise output in the form of a JSON object or dictionary, with key-value pairs organised in a tabular style, thereby enabling straightforward and reliable verification. Solving these tasks requires agents to formulate plans, decompose problems into sub-steps, select and use appropriate external tools (e.g., document processors, code interpreters), and integrate intermediate results into a final solution.

We now describe the process of constructing DEEPSYNTH. We first outline the criteria for our tasks, then describe our data collection pipeline, and conclude with an analysis of the collected data.

### 2.1 CRITERIA FOR DEEPSYNTH TASKS

Motivated by prior benchmarks (Mialon et al., 2023; Yoran et al., 2024; Wei et al., 2025; Phan et al., 2025), the design of DEEPSYNTH tasks is driven primarily by criteria that promote the future development of Agents' information seeking and synthesis capabilities towards practical and grounded goals. Specifically, our criteria consist of:

a) **Multi-source Information Synthesis**: Tasks should require agents to identify connections across multiple data sources or to combine information from them to produce a coherent solution. More specifically, tasks are designed such that agents must not only fetch relevant information but also perform subsequent operations on it (see Table 1).

b) **Inspired by the Real World**: Experts were instructed to draw inspiration from real-world situations. The tasks are designed so that any resulting insights would conceivably be able to shape the decisions and actions of an individual or a group of people, such as *political scientists, policy makers, travel agents, etc.*

c) **Verifiable Answers**: A task that has a closed-form answer, which can be automatically verified and is stable over time, making it suitable for reproducible evaluation. While the answers to our tasks may be better suited to open-form answers, properly argued and grounded in citations, we necessarily restrict them to maintain their verifiability.

d) **Diversity**: Our benchmark is designed to span a wide range of tasks, requiring agents to gather and reason over information across 67 countries and 7 distinct domains. Beyond geographic and topical diversity, the tasks also encompass temporal analyses, comparative evaluations across groups or categories, and relational reasoning, ensuring that agents are tested on a variety of reasoning modes.

e) **Robustness Against Memorisation**: Similar to Mialon et al. (2023), we ensured that the tasks are explicitly constructed to mitigate data contamination and prevent superficial memorisation. The gold-standard answers are intentionally built to be non-retrievable through verbatim lookup in known pre-training corpora or direct web search, compelling systems to plan and perform multi-step reasoning to derive the correct output.

### 2.2 DATA COLLECTION

A common practice in designing deep agentic benchmarks is to start with a fact and then craft a question from it, making the answer difficult to locate (Wei et al., 2025). Since our goal was to ensure answers are non-retrievable, we adopted a different approach. Building DEEPSYNTH involved four

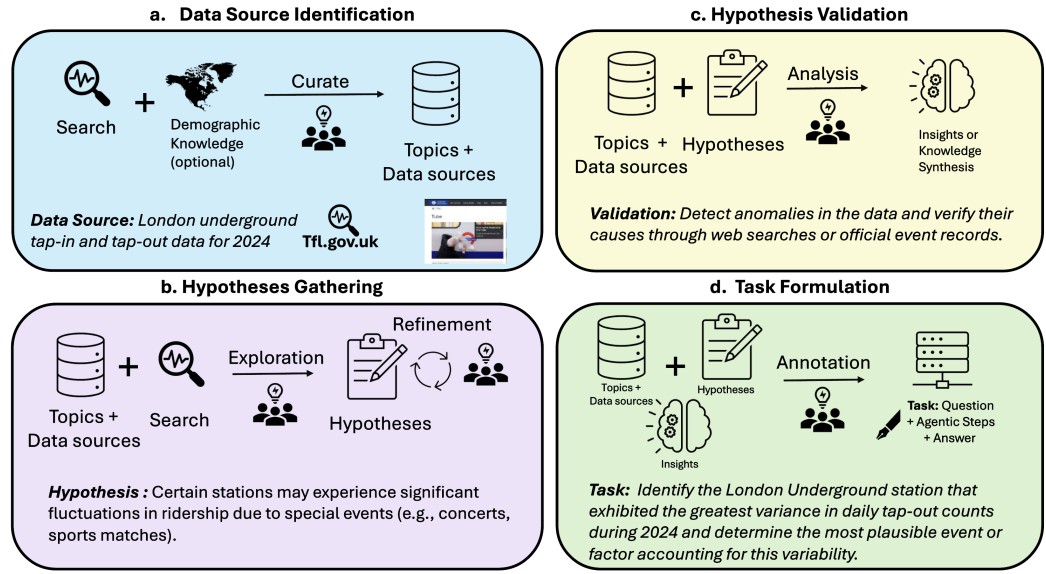

Figure 2: An overview of our data collection process for building the DEEPSYNTH benchmark

key steps: (a) identifying data sources, (b) gathering hypotheses, (c) performing analyses, and (d) formulating tasks (see Figure 2).

**Data Source Identification.** In this step (see Figure 2, left), we engaged 16 human experts[2] to propose a diverse set of data sources and topics, drawing on their expertise, demographic backgrounds, and interests. Given the complexity of the annotation process and the need for efficient coordination, participation was restricted to individuals with whom we maintained direct communication, along with the paper's core authors. We collected 223 data sources across 7 domains (*socio-economic, finance, environment, science, education, transportation, political/socio-political*). We excluded data sources that originated from untrustworthy or non-official websites, including those requiring user authentication, as well as sources containing information that contradicted other verified references. Our objective was to curate tasks that are useful to individuals or groups; therefore, we filtered data sources to retain only those from which useful, verifiable insights could be drawn. For example, we included official statistical reports on *"the gender gap in labour force participation rates in Australia"*, *"computer and digital literacy rates in Sri Lanka"*, and *"air quality and pneumonia-related deaths across regions in the UK"*, since such data enables clear downstream reasoning tasks (e.g., analyzing temporal trends, comparing across regions, or correlating with policy interventions).

**Hypothesis Generation.** We then asked annotators to formulate one or two hypotheses—plausible insights that could be inferred from the selected data sources (see Figure 2 bottom left). The objective of this step was to elicit hypotheses that are both insightful and practically valuable, encouraging reasoning beyond surface-level fact retrieval (see § 2.1(e)). For example, one such hypothesis was: *"Is there a linear relationship between air quality and pneumonia-related deaths across regions in the UK?"*. Data sources that did not meet the criteria of *usefulness* and *insightfulness* (see § 2.1(b)) were subsequently filtered out.[3] After this step, we retained a total of 155 data sources, each paired with its corresponding set of hypotheses.

**Hypothesis Validation.** In this step, annotators were tasked with conducting a detailed analysis of each data source to assess the validity of the proposed hypotheses (see Figure 2 top right). The objective was twofold: (i) to verify whether the data supported the hypotheses, and (ii) to derive tasks with *verifiable* answers (see § 2.1(c)). Hypotheses that failed to meet the verifiability criterion were

---

[2]Details about the annotators are provided in Appendix A.2.

[3]Please note this step involves a degree of subjectivity, and we relied on the domain knowledge and judgment of our annotators to ensure the quality of the retained data sources and hypotheses.

| Statistic | Value |
|---|---|
| Total Tasks | 120 |
| Avg. Question tokens | 78.49 |
| Avg. Intermediate steps | 7.54 |
| Web pages per Task | 4.2 |
| Avg. Annotation Time | 5.5 hours |
| **Synthesis Operations** | **%** |
| Trend Detection | 20.93% |
| Average | 11.05% |
| Correlation | 6.98% |
| Ranking | 19.77% |
| Anomaly Detection | 6.98% |
| Counting and Comparing | 33.72% |
| Filtering | 0.58% |

Table 1: DEEPSYNTH statistics across tasks.

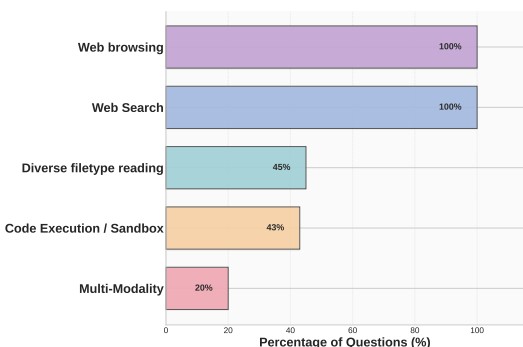

Figure 3: Percentage of tasks per capabilities required to solve DEEPSYNTH.

refined or discarded. Following this validation and filtering process, we retained 130 data sources, each paired with its corresponding, verified hypothesis.

**Task Formulation.** Finally, annotators were asked to formulate task questions along with intermediate steps, supporting evidence and corresponding answers. We note that the intermediate steps indicate only one possible reasoning path or planning from question to answer, and that no model or agent necessarily needs to imitate that path. Since DEEPSYNTH tasks often rely on multiple pieces of supporting evidence and reference various documents or tables, annotators were instructed to provide the URLs where the data sources can be accessed. Additionally, they were asked to include a brief explanation of how the task can be solved, specifying any tools, code snippets, or mathematical formulas used in the solution. We provide more examples and additional statistics in §A.3.

**Data Validation.** All questions went through a second annotation stage, where another annotator independently answered the question. Only tasks where the answers from both annotators were identical were retained in the dataset, leaving finally 120 challenging information synthesis tasks.

## 2.3 DATA STATISTICS

Table 1 summarises the key statistics of our benchmark. Tasks in DEEPSYNTH are highly detailed, with an average length of $78.49$ tokens, an average of $7.54$ intermediate reasoning steps and requiring navigation through an average of $4.2$ web pages to reach a solution. Additionally, on average, formulating each task (from data source identification to task formulation) took the annotators approximately 5.5 hours. This number highlights the challenge in creating such tasks. Overall, all these numbers underscore the inherent complexity and challenge of the benchmark. Moreover, the tasks encompass a diverse range of analytical and reasoning skills, including correlation analysis, anomaly detection, and identification of causal or linear relationships — as reflected in Table 1. Table 5 presents the regions covered by our benchmark, along with the percentage of tasks corresponding to each region. Notably, the benchmark comprises a higher proportion of tasks from Europe and Asia, with some tasks spanning multiple countries and regions. Figure 3 provides an overview of the capabilities required by agents to solve the benchmark and their prevalence in tasks. In particular, we observe that web search and browsing are the most critical skills for retrieving the correct information.

## 3 EVALUATION SETUP

**Models.** We use DEEPSYNTH to benchmark five state-of-the-art models: (a) o4-mini (OpenAI, 2025c), (b) GPT-4.1 (OpenAI, 2024), (c) GPT-5 (OpenAI, 2025b), (d) Gemini-2.5-Pro (Comanici et al., 2025) and (e) DeepSeek-R1 (Guo et al., 2025). For Gemini-2.5-Pro, we use "dynamic thinking", where the model decides how much to think. GPT-5 was evaluated using "high reasoning effort".

| Model | F1 Score | Precision | Recall | Exact Match | LLM Judge Score |
|---|---|---|---|---|---|
| **LLM Baselines** | | | | | |
| o4-mini (2025-08) 🔒 | 3.05 | 2.33 | 4.39 | 0.0 | 0.0 |
| GPT-4.1 (2025-08) 🔒 | 3.46 | 2.86 | 4.39 | 0.0 | 0.0 |
| o3 (2025-08) 🔒 | 3.29 | 2.85 | 3.90 | 0.0 | 0.0 |
| GPT-5.1 (2025-08) 🔒 | 3.83 | 2.98 | 5.37 | 0.0 | 0.0 |
| Gemini-Pro-2.5 (2025-08) 🔒 | 6.25 | 4.71 | 9.27 | 0.0 | 5.0 |
| GPT-5.2-Pro (2026-02) 🔒 | 8.70 | 8.45 | 8.96 | 6.25 | 6.67 |
| DeepSeek-R1-Chat (2025-08) 🔓 | 3.23 | 2.75 | 3.90 | 1.67 | 2.5 |
| DeepSeek-R1-Reasoner (2026-02) 🔓 | 2.80 | 2.73 | 2.87 | 2.50 | 6.67 |
| **Framework-based Agents** | | | | | |
| o3-deep-research (2025-08) 🔒 | 8.97 | 7.73 | 10.69 | 2.50 | 17.5 |
| Smolagent (GPT-4.1) 🔓 | 3.75 | 3.27 | 4.39 | 2.50 | 7.5 |
| Smolagent (GPT-5) 🔓 | 6.42 | 6.34 | 6.50 | 1.67 | 2.5 |
| OWL (GPT-4.1) 🔓 | 5.41 | 4.62 | 6.52 | 1.67 | 12.5 |

Table 2: Performance comparison on the DEEPSYNTH benchmark (Pass@1). **F1, Precision, Recall and Exact Match** measure the quality of model predictions. **LLM Judge (%)** reports the average precision. Models with 🔒 are models or framework which are closed, while 🔓 are open-source.

We also investigate the performance of three state-of-the-art (deep research) agentic frameworks: (a) o3-deep-research (OpenAI, 2025a); (b) smolagents (Roucher et al., 2025), which is a minimalist framework focused on simplicity and rapid prototyping. It uses a standard ReAct loop (Yao et al., 2023) and its primary distinguishing feature is that it expresses all actions, such as tool use, as code, which is parsed out of the response and executed Wang et al. (2024); (c) OWL (Hu et al., 2025), which employs a role-playing strategy where a planner and an executor collaboratively solve tasks, optionally augmented with smaller, more specialised 'workers'. Both OWL and smolagents have been open-sourced. The details of their tool capabilities are listed in Table 10. All models were prompted using the same instructions, provided in Appendix A.5.

**Metrics.** All tasks require models to generate outputs in a JSON format (or lists of JSON objects). Our strictest metric is *exact match*, meaning that all keys and values must be correct. For partial evaluation, we check how many *key-value pairs* are correct (out of the total pairs) and report precision, recall and F1-score. As an additional 'soft' metric, we follow Wolfson et al. (2025) and leverage the LLM-as-a-judge (with an identical prompt, see Fig. 10) reporting average precision. This serves two purposes: 1) for small string differences (with semantic equivalence), this method will reward answers beyond exact match, 2) in case of numerical answers, a small margin (1% to 5.5% difference) can still be considered correct, hence providing more granular and permissible scores.

## 4 RESULTS

### 4.1 MAIN RESULTS

Table 2 shows the performance of SOTA models on DEEPSYNTH. We first evaluate the parametric knowledge and reasoning capabilities of LLMs. The results show that GPT-5.2-Pro achieves the highest F1 score of $8.70$, and both GPT-5.2-Pro and DeepSeek-R1-Reasoner achieve the highest LLM Judge score of $6.67$, indicating substantial room for improvement. Interestingly, the performance gap between reasoning models (e.g. Gemini-2.5-Pro, GPT-5.1, DeepSeek-R1) and general-purpose LLMs (e.g., GPT-4.1) is relatively small on F1 score. This finding suggests that the key bottleneck lies not in reasoning ability alone, but in the availability of the necessary information for reasoning. We investigate this observation in greater depth in Analysis; see §5. Further, under the strict exact-match metric, we observe that almost all models obtain a score of zero, indicating that none can solve even a single task perfectly. The poor performance of base LLMs indicates that internal retrieval of parametric knowledge is insufficient, showing that these tasks are robust against memorisation (see criteria §2.1(e)). This also highlights the need to augment these models with external tools.

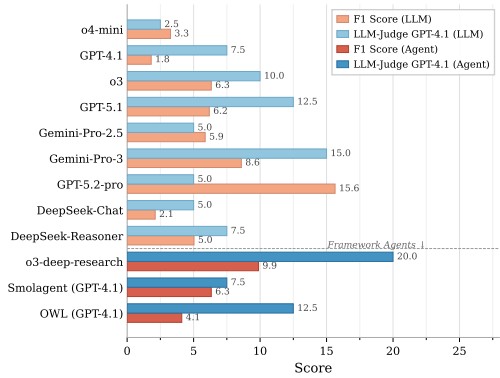

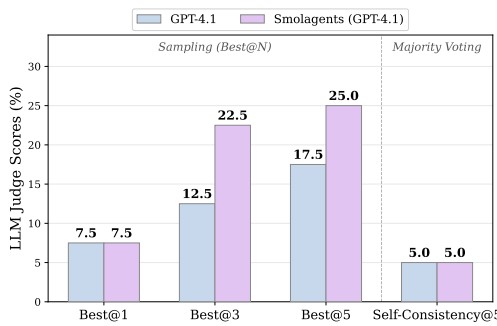

(a) Pass@1 performance on DEEPSYNTH-Dev.

(b) Best-of-N performance on DEEPSYNTH-Dev.

Figure 4: Performance comparison on the DEEPSYNTH-Dev benchmark. **(a) F1 Score** measures the quality of model predictions. **LLM-Judge (GPT-4.1)** reports the average precision as judged by GPT-4.1. Light bars denote LLM baselines; dark bars denote framework-based agents. **(b)** Best-of-N LLM Judge Scores comparing GPT-4.1 standalone vs. Smolagents (GPT-4.1) across $N \in \{1, 3, 5\}$ and and Self-Consistency@5 (majority voting).

To investigate this further, we evaluated our benchmark using three agentic frameworks that integrate external tools, including simulated web browsing, web search, and a code interpreter. We find that o3–deep-research, which incorporates web search and an executable code interpreter, outperforms the base o3 model by $5.68$ F1 score and $2.50$ EM. Furthermore, smolagents and OWL achieve some gains, with improvements of $0.29$ and $1.95$ F1 points and $2.5$ and $1.67$ EM points respectively, over the base GPT-4.1. Overall, we observe that all systems perform poorly. These findings emphasise that effectively solving tasks in DEEPSYNTH requires enhanced tool-use capabilities. Interestingly, we find that low precision indicates that all models frequently produce incorrect or extraneous answers.

**DEEPSYNTH-Dev Results.** Figure 4a presents results on the DEEPSYNTH-Dev subset. Among standalone LLMs, GPT-5.2-Pro achieves the highest F1 score (15.6), while Gemini-Pro-3 leads on the LLM-Judge metric (15.0), suggesting it produces semantically reasonable outputs that strict matching penalizes. Among agents, o3-deep-research attains the highest LLM-Judge score (20.0), reinforcing that tool augmentation benefits synthesis-heavy tasks. We observe a consistent gap between LLM-Judge and F1 scores across all models. Our manual evaluation suggests that this discrepancy primarily arises from failures to produce numerically precise or structurally exact outputs.

**Best-of-N and Self-Consistency Analysis.** Figure 4b examines whether multiple attempts improve task completion on DEEPSYNTH-Dev. Under Best@5, Smolagents (GPT-4.1) reaches 25.0% LLM-Judge accuracy compared to 17.5% for GPT-4.1, suggesting that tool-use introduces beneficial variance across runs. However, self-consistency (majority voting at $N$=5) yields only 5% accuracy for both systems, with low average consistency scores (0.27), indicating that correct answers rarely emerge as the majority prediction. The stark contrast between Best@5 and self-consistency (27.5% vs. 5% for Smolagents) demonstrates that current agents exhibit high output variance on DEEPSYNTH tasks. Occasional runs succeed, but models lack the reliability needed for consistent, correct answers.

**Ablation Study.** To assess the role of different capabilities on DEEPSYNTH, we perform an ablation study. As shown in Table 3 (top), performance shows consistent declines across all metrics when any capability is removed, with the largest drop (1.81 F1 points) observed when search is excluded. While the overall changes are modest due to the low baseline performance, these trends indicate that document processing, code execution, and search each contribute to task success, highlighting the multifaceted challenges posed by DEEPSYNTH.

| Model | Performance Metrics | | | |
|---|---|---|---|---|
| | **F1** | **Precision** | **Recall** | **EM** |
| **OWL (Full)** | 5.41 | 4.62 | 6.52 | 1.67 |
| **Tool Ablation** | | | | |
|    − Web Browsing Toolkit | 4.80 | 4.20 | 5.60 | 1.67 |
|    − Search Toolkit | 3.60 | 2.96 | 4.61 | 0.0 |
|    − Document Processing Toolkit | 4.90 | 4.50 | 5.4 | 1.67 |
|    − Code Execution Toolkit | 4.82 | 4.30 | 5.50 | 0.0 |
| **Reasoning Chain Ablation** | | | | |
| **GPT-4.1** | 3.46 | 2.86 | 4.39 | 0.0 |
|    + Intermediate Step | 9.36 | 8.76 | 10.05 | 5.0 |
| **Smolagent (GPT-4.1)** | 3.75 | 3.27 | 4.39 | 2.50 |
|    + Intermediate Step | 10.50 | 8.96 | 12.70 | 10.0 |

Table 3: Ablation Study. **Tool Ablation:** Comparing the benefits of using different tools on DEEP-SYNTH. **Reasoning Chain Ablation:** Studying the role of planning given the intermediate steps.

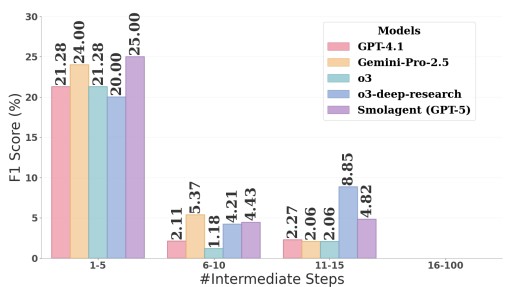

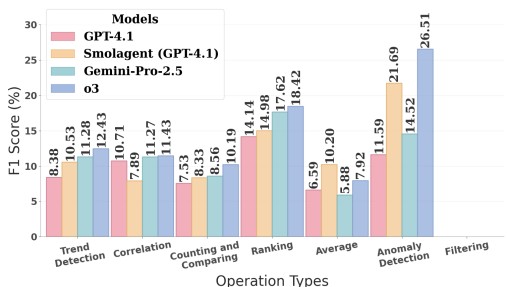

Figure 5: F1-scores across intermediate steps.  Figure 6: F1-scores across synthesis operations.

## 5 ANALYSIS

In order to understand the challenges of solving DEEPSYNTH questions, we analyse performance across different data collection strategies, followed by a qualitative analysis of model errors.

**RQ$_1$: How do models perform as the number of intermediate steps increases?** We break down the models' performance based on the number of intermediate steps entailed by DEEPSYNTH tasks. Figure 5 presents the performance breakdown, highlighting that all models struggle as the number of intermediate steps increases, which can be considered an indicator of the task's complexity. Notably, the agentic frameworks (o3-deep research and smolagents + GPT-5) perform better for 11-15 intermediate steps, while they are on par with other LLMs for smaller numbers of intermediate answers. Given that tasks in DEEPSYNTH require an average of 7.54 intermediate steps, these results provide insights into why the benchmark is so challenging.

**RQ$_2$: Does providing agents with intermediate steps improve their performance?** We evaluate how agents perform when they are provided with the ground truth intermediate reasoning steps (i.e. planning) without revealing the intermediate answers. As shown in Table 3, model performance improves substantially under this setting, with GPT-4.1 and smolagents + GPT-4.1 showing large gains. Both EM and F1 scores increase, indicating that models appear to lack planning capabilities.

**RQ$_3$: Which synthesis operations are more challenging?** To further assess the models' analytical capabilities, we examine their performance on different synthesis operations when intermediate steps are provided alongside the task questions (see Table1, 11). Figure 6 presents the results across various operation types, revealing substantial variation in task-specific performance. More specifically, we observe that o3 model achieves the highest F1 score in anomaly detection (26.51%), significantly

| Region | % of Tasks | GPT-4.1 | o3-deep-research | Gemini-2.5-Pro | smolagents |
|---|---|---|---|---|---|
| Africa | 8.3% | 0.0 | 0.0 | 0.0 | 0.0 |
| North America | 11.7% | 4.65 | 8.00 | 12.00 | 8.33 |
| South America | 5% | 0.0 | 25.00 | 0.0 | 0.00 |
| Asia | 29.2% | 3.36 | 12.70 | 6.50 | 11.88 |
| Europe | 38.3% | 3.45 | 10.83 | 4.91 | 5.28 |
| Oceanic | 10.8% | 8.96 | 14.43 | 6.67 | 24.00 |
| Others | 12.5% | 3.12 | 6.45 | 12.12 | 0.0 |

Table 5: **Multi-Regional Analysis**: Agent performance across region-specific tasks (**F1 score**). NOTE: A question may span multiple regions. "Others" contains tasks without regional association.

outperforming the other agents, while Gemini-2.5-Pro and smolagents + GPT-4.1 exhibit moderate gains over GPT-4.1 across most task categories. Trend detection and ranking also demonstrate relatively strong performance for Gemini-2.5-Pro and o3, indicating that these models can effectively capture certain structured patterns. In contrast, none of the models exhibit measurable performance on filtering tasks, which may partly reflect the limited number of filtering tasks in the benchmark (see Table 1). Overall, these findings suggest that, although some agents can successfully identify anomalous or structured patterns, significant improvements are required for tasks involving arithmetic, comparative reasoning, or complex multi-step analysis.

| Error Cause | No. of instances |
|---|---|
| Navigation Error | 15 |
| No answer | 4 |
| Technical Issue | 4 |
| Synthesis Error | 16 |

Table 4: Error analysis for OWL (GPT-4.1). Navigation and synthesis errors are the most prominent.

**RQ$_4$:** **What types of errors do models commonly make?** To better understand the challenges in solving DEEPSYNTH, we manually analysed a random subset of 32 tasks[4] in which OWL + GPT-4.1 made errors[5]. We focus on OWL because, as an open-source framework, it enables detailed examination of execution traces and interactions between agents and tools. We categorize errors into four types, with their frequencies summarized in Table 4: (1) *Navigation errors* – when the agent fails to locate or access the correct source of information, such as navigating to the wrong web page, document, or section; (2) *No Answer* – when the agent does not respond or fails to generate any output; (3) *Technical Issue* – errors caused by system limitations, software bugs, or tool malfunctions that prevent task completion, independent of reasoning or navigation; and (4) *Synthesis Error* – when the agent reaches an incorrect conclusion despite accessing the correct information, due to flaws in logical reasoning, interpretation, or multi-step analytical processes.

This analysis is multi-label, as a single instance may exhibit multiple error types. The majority of errors—15/32 due to navigation and 16/32 due to reasoning—highlight that DEEPSYNTH presents significant challenges even for state-of-the-art open-source models. Figure 9 illustrates a failure case of OWL, in which the correct URL was found, but the agent fails to interact correctly with the website and its database interface.

**RQ$_5$:** **How do agents perform on tasks from different regions?** We observe that o3-deep research exhibits the most consistent cross-regional capability, particularly in the high-volume areas such as Europe and Asia. Notably, all models fail on Africa-related tasks, achieving an F1 score of 0.0. These findings highlight the presence of strong geographical biases in current models and suggest that their performance is not globally uniform, likely reflecting imbalances in the distribution and coverage

---

[4]Subset chosen due to the time and cost of manually analysing all outputs.

[5]Two annotators who were not involved in the original data annotation conducted this analysis.

| Dataset | Real World | Multi Regional | Information Synthesis | Multi-Part Answers |
|---|---|---|---|---|
| GAIA (Mialon et al., 2023) | partial | ✗ | partial | ✗ |
| AssistantBench (Yoran et al., 2024) | ✓ | ✗ | ✗ | partial |
| BrowseComp (Wei et al., 2025) | ✗ | ✗ | ✗ | ✗ |
| HLE (Phan et al., 2025) | partial | partial | partial | ✗ |
| **DEEPSYNTH** | ✓ | ✓ | ✓ | ✓ |

Table 6: Comparison of datasets on various reasoning and retrieval capabilities.

of their training data. Since DEEPSYNTH contains a diverse set of tasks from multiple regions, it naturally increases the overall difficulty of the benchmark.

# 6    RELATED WORK

As LLM-based agents improve in reasoning, tool usage, and interaction across diverse environments, researchers have sought to evaluate LLMs on questions that require multi-hop reasoning skills (Wu et al., 2025; Wolfson et al., 2025), code generation (Jimenez et al., 2024; Chan et al., 2024; Ouyang et al., 2025; Starace et al.), information seeking (Wei et al., 2025; Yoran et al., 2024) and even general assistance capabilities (Mialon et al., 2023). Table 6 summarises the key differences among popular agentic benchmarks; most of these correspond to the criteria we considered in § 2.1 for DEEPSYNTH tasks' design. A distinctive feature of DEEPSYNTH is its multi-part answers, where each response comprises multiple components—e.g., a JSON object containing event causes (strings), percentages (floats), dates or years (integers)—with explicit logical links (e.g., key-value pairs). This structure makes the benchmark particularly challenging, as it requires agents to retrieve, reason, and integrate heterogeneous information correctly.

Several existing benchmarks share partial overlap with DEEPSYNTH but lack a systematic evaluation of information synthesis. For instance, GAIA (Mialon et al., 2023) requires planning and information seeking but involves limited synthesis and less realistic tasks. BrowseComp (Wei et al., 2025) is an information-seeking benchmark comprising challenging, invertedly constructed questions that require persistent, multi-hop web navigation to uncover hard-to-find facts; in contrast, DEEPSYNTH moves beyond retrieval to systematically evaluate information synthesis through multi-part, structured answers. AssistantBench (Yoran et al., 2024) addresses real-world gaps and includes limited multi-part answers, but omits other essential aspects. Humanity's Last Exam (Phan et al., 2025) offers precise, unambiguous, and non-searchable questions, yet these are often obscure and detached from real-world contexts. In contrast, DEEPSYNTH is, to the best of our knowledge, the first benchmark to systematically evaluate information synthesis across realistic, multi-step tasks.

# 7    CONCLUSION

We presented DEEPSYNTH bench, a new benchmark comprising 120 challenging and diverse tasks across 67 countries. By combining planning, tool use, and multi-step reasoning, DEEPSYNTH aims to evaluate the ability of agents to move beyond shallow retrieval and engage in goal-directed, information-rich problem solving. DEEPSYNTH was inspired by real-world problems, and its tasks were designed to be strictly verifiable, geopolitically diverse, and robust against memorisation. Our experiments demonstrated the difficulty of our benchmark, with both state-of-the-art LLMs and specialized deep research agents struggling to solve any significant number of tasks. The best of the former (Gemini-Pro-2.5) achieved an F1 score of only $6.25$ with no task reaching a perfect score, while the best of the latter (o3-deep-research) reached $8.97$. These results help establish that there is substantial room for improvement on multi-source information synthesis, and we hope DEEPSYNTH will inspire future work, starting with improving navigation and synthesis, and addressing the significant geopolitical biases we observed.

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

# A APPENDIX

## A.1 MORE RESULTS AND ANALYSIS

This section collects some additional results and analysis. Specifically:

Table 7 shows how long was required for state-of-the-art LLMs and specialized deep research agents to run DEEPSYNTH bench.

Table 8 presents the intermediate step accuracy and error propagation on the DEEPSYNTH-Dev tasks.

Table 9 summarises the cost and output token characteristics of the models evaluated in our experiments. For models that produce structured multi-stage reasoning traces, we report both reasoning-token ranges and final completion-token ranges. Costs correspond to the total API price per full run of a DEEPSYNTH task.

Table 10 presents a brief comparison across the Agentic Framework tool capabilities of the specialized deep research agents we apply to DEEPSYNTH bench (OpenAI, 2025a; Roucher et al., 2025; Hu et al., 2025).

Table 11 provides definitions and examples on the key information synthesis operations in DEEP-SYNTH's tasks.

Table 12 collects F1-score, Precision, Recall and EM scores to highlight the role of planning intermediate steps on LLM models.

Table 13 presents the performance comparison on the DEEPSYNTH-Dev benchmark (Pass@1).

Finally, Figure 9 shows an example run using the OWL framework, and illustrates errors when trying to collect and reason about data.

| Model | Avg. Time (sec.) |
|---|---|
| o4-mini (2025-08) | 20.8 |
| GPT-4.1 (2025-08) | 7.52 |
| GPT-5 (2025-08) | 83.41 |
| Gemini-Pro-2.5 (2025-08) | 34.10 |
| DeepSeek-R1 (2025-08) | 5.2 |
| o3-deep-research (2025-08) | 645.39 |
| Smolagent (GPT-4.1) | 35.84 |
| OWL (GPT-4.1) | 1025.5 |

Table 7: Average Time per instance to run our benchmark

**Process Evaluation**   Recently, several works have shown that LLMs make mistakes in their intermediate reasoning steps (Paul et al., 2024a; Mondorf & Plank, 2024; Lee & Hockenmaier, 2025) and can be unfaithful to their own reasoning (Paul et al., 2024b). Gritta et al. (2026) argued that outcome-only metrics are insufficient for critical applications and proposed compliance checklists to verify that agents follow recommended protocols. Motivated by these findings, we evaluate intermediate step accuracy by requiring models to emit structured outputs after each decomposition sub-step. While end-to-end F1 scores capture overall performance, they obscure *where* in the reasoning chain failures originate and whether errors at early stages propagate to corrupt downstream computation. We evaluate three models: GPT-4.1, DeepSeek-R1, and GPT-5.2 on 40 DEEPSYNTH tasks, scoring each intermediate output against its corresponding gold answer using entity-normalized F1 with format normalization to handle structural variation between predicted and gold JSON outputs. GPT-4.1 and DeepSeek-R1 operate without web access as instruction-following models, while GPT-5.2 operates with web search enabled via the Responses API.

Table 8 presents per-step F1 and error propagation rates. All three models exhibit a steep accuracy decay: retrieval steps (steps 1–2) achieve 2–12% F1, indicating partial but incomplete data acquisition, while computation and reasoning steps (steps 3+) collapse to near zero. Error propagation is near-total—when a step fails (F1 < 50), the subsequent step also fails 91–100% of the time, with recovery

|          | DeepSeek-R1 | GPT-4.1 | GPT-5.2 | Avg. Prop. (%) |
|----------|-------------|---------|---------|----------------|
| Step 1   | **11.2**    | 10.0    | 4.1     | —              |
| Step 2   | **12.4**    | 9.8     | 2.6     | 97.0           |
| Step 3   | **3.9**     | 3.3     | 0.5     | 100.0          |
| Step 4   | 1.4         | **2.4** | 0.0     | 100.0          |
| Step 5   | 0.2         | 0.0     | 0.0     | 100.0          |
| Step 6   | 0.0         | 0.0     | 0.0     | 100.0          |
| **Final Answer** | **20.1** | 18.5 | 16.7 | —              |

Table 8: **Intermediate step accuracy and error propagation.** Per-step F1 (%) on 40 DEEPSYNTH tasks. *Prop.* denotes error propagation rate: the percentage of cases where failure at step $k$ (F1 $< 50$) also results in failure at step $k+1$. Steps with $n < 3$ are omitted. GPT-4.1 and DeepSeek-R1 operate without web access; GPT-5.2 operates with web search enabled. All models show steep accuracy decay and near-total error propagation.

rates below 10%. This confirms that the low end-to-end F1 scores reported in Table 2 are driven primarily by early retrieval failures that cascade irrecoverably through the reasoning chain.

A notable divergence emerges between models. Among instruction-following models, DeepSeek-R1 achieves the highest intermediate accuracy at early steps (11.2% at step 1 vs. 10.0% for GPT-4.1) and the highest final-answer F1 (20.1%), suggesting that its extended chain-of-thought reasoning provides a modest advantage in producing more accurate intermediate data. However, GPT-5.2 presents a strikingly different profile: despite having access to web search, it achieves the *lowest* intermediate step accuracy (4.1% at step 1) while maintaining comparable final F1 (16.7%). Qualitative analysis of GPT-5.2's intermediate outputs reveals the cause of this apparent paradox. Rather than producing approximate values, GPT-5.2 performs genuine web search: correctly identifying target databases, API endpoints, indicator codes, and download URLs, but reports structured failures when its execution environment cannot process the retrieved sources (e.g., JavaScript-rendered tables, binary `.xlsx` files, or bulk data downloads requiring interactive queries).

| Model          | Cost (USD) | Output Tokens (Min / Max)                        |
|----------------|------------|--------------------------------------------------|
| deepseek-R1    | $0.041     | 687.4 (avg)                                      |
| GPT-4          | $0.432     | 188 (min) / 383 (max)                            |
| GPT-5          | $5.52      | 1600–15872 (reasoning), 201–425 (completion)     |
| o4-mini        | $1.39      | 448–5760 (reasoning), 40–392 (completion)        |
| o3 deep-research | $184.61  | 448–5760 (reasoning), 40–392 (completion)        |
| Gemini-Pro-2.5 | $11.78     | 12995 (avg)                                      |

Table 9: Model cost and output token ranges.

| Framework                | Code Interpreter | Web Search | API Calls | Document Processing | Browser Simulation |
|--------------------------|------------------|------------|-----------|---------------------|--------------------|
| o3-deep-research (2025-08) | ✓              | ✓          | ✗         | ✗                   | ✗                  |
| Smolagent (GPT-5)        | ✓                | ✓          | ✗         | ✗                   | ✗                  |
| OWL                      | ✗                | ✓          | ✓         | ✓                   | ✓                  |

Table 10: Agentic Framework Tool Capabilities Comparison. **Note:** ✓ indicates capability present; ✗ indicates capability not available. Web Browser functionality is included within Web Search capabilities for applicable frameworks.

| Operation | Description | Example |
|---|---|---|
| Trend Detection | Identifying patterns, directions, or changes in data over time or across contexts. | Climate events over the last decade to detect a trend of increasing wildfires. |
| Average | Determining a representative value that summarises numerical data collected from multiple sources. | Synthesising daily temperatures from multiple weather stations to report the average regional temperature. |
| Correlation | Measuring relationships or associations between two or more variables. | Synthesising data from health studies to find correlations between exercise frequency and cholesterol levels. |
| Ranking | Ordering items or facts based on specific criteria or importance. | Compiling product reviews from different websites and ranking products by overall rating. |
| Anomaly Detection | Identifying data points or patterns that significantly deviate from the norm. | Synthesising sensor data from multiple factories to detect unusual machine behaviour. |
| Counting and Comparing | Quantifying occurrences and comparing values across sources or categories. | Counting positive vs negative mentions of a policy in news articles and comparing the proportions. |
| Filtering | Selecting relevant information based on criteria, quality, or thresholds. | Selecting only recent when synthesising research on climate change. |

Table 11: Key operations in information synthesis, their definitions, and examples of application.

| Model | Performance Metrics | | | |
|---|---|---|---|---|
| | F1 | Precision | Recall | EM |
| **o3** | 3.29 | 2.85 | 3.90 | 0.0 |
| + Intermediate Step | 12.87 | 11.38 | 14.81 | 7.50 |
| **Gemini-Pro-2.5** | 6.25 | 4.71 | 9.27 | 0.0 |
| + Intermediate Step | 10.40 | 8.36 | 13.76 | 7.50 |

Table 12: **Analysis.** Studying the role of planning/intermediate steps.

## A.2 DATACARD AND ANNOTATION GUIDELINES

DEEPSYNTH currently spans 67 unique countries, including Afghanistan, Algeria, Australia, Austria, Belgium, Bhutan, Brazil, Brunei, Cambodia, Cameroon, Canada, Chile, China, Czech Republic, Denmark, Estonia, Fiji, Finland, France, Germany, Ghana, Greece, Greenland, Hungary, Iceland, India, Indonesia, Italy, Japan, Laos, Latvia, Lebanon, Liechtenstein, Lithuania, Luxembourg, Malaysia, Maldives, Myanmar, Nepal, Netherlands, New Zealand, Nigeria, Norway, Pakistan, Peru, Philippines, Poland, Portugal, Qatar, Singapore, Slovakia, South Africa, South Korea, Spain, Sri Lanka, Sweden, Switzerland, Taiwan, Tajikistan, Thailand, Tunisia, Turkmenistan, United Kingdom, United States, Uzbekistan, Vietnam, and Zimbabwe.

**Annotator Demographics.** We provide additional information that may be relevant for analysing this dataset. Building DEEPSYNTH required the work of expert annotators, who devised the task questions and their answers, and who independently annotated the questions to assess their non-ambiguity. We have **81.25**% of the annotator PhD holders. Both come from the following population:

1. **Age**:
   (a) 18 - 25 : 12%
   (b) 26 - 35 : 68%
   (c) 36 - 45 : 18%

| Model | F1 Score | Precision | Recall | Exact Match | LLM Judge Score (GPT-4.1) |
|---|---|---|---|---|---|
| **LLM Baselines** | | | | | |
| o4-mini (2025-12) 🔒 | 3.26 | 2.63 | 4.29 | 0.0 | 2.5 |
| GPT-4.1 (2025-12) 🔒 | 1.81 | 1.57 | 2.14 | 0.0 | 7.5 |
| o3 (2025-12) 🔒 | 6.33 | 5.68 | 7.14 | 1.25 | 10.0 |
| GPT-5.1 (2025-12) 🔒 | 6.18 | 6.72 | 5.71 | 1.25 | 12.5 |
| Gemini-Pro-2.5 (2025-12) 🔒 | 5.87 | 4.98 | 7.14 | 0.0 | 5.0 |
| Gemini-Pro-3 (2025-12) 🔒 | 8.59 | 7.53 | 10.0 | 2.5 | 15.0 |
| GPT-5.2 (2025-12) 🔒 | 15.64 | 18.45 | 13.57 | 2.5 | 5.0 |
| DeepSeek-Chat (2025-12) 🔓 | 2.11 | 2.08 | 2.14 | 0.0 | 5.0 |
| DeepSeek-Reasoner (2025-12) 🔓 | 5.03 | 4.49 | 5.71 | 1.25 | 7.5 |
| **Framework-based Agents** | | | | | |
| o3-deep-research (2025-08) 🔓 | 9.88 | 7.55 | 14.29 | 7.50 | 20.0 |
| Smolagent (GPT-4.1) 🔓 | 6.33 | 5.68 | 7.14 | 7.14 | 7.5 |
| OWL (GPT-4.1) 🔓 | 4.11 | 3.95 | 4.29 | 0.0 | 12.5 |

Table 13: Performance comparison on the DEEPSYNTH-Dev benchmark (Pass@1). **F1, Precision, Recall and Exact Match** measure the quality of model predictions. **LLM Judge (%)** reports the average precision. Models with 🔒 are models or framework which are closed, while 🔓 are open-source.

2. **Gender**: 25% Female, 75% Male

3. **Nationality**: India, Greece, Luxembourg, Slovakia, UK, China, Peru, Romania, Turkey, Kosovo, Germany

4. **Academic Background**:

   (a) Bachelor's Degree: 6.25%

   (b) Master's Degree: 12.5 %

   (c) PhD : **81.25%**

The guidelines that were given to the annotators are presented in Figures 7 and 8. The goal of this benchmark is to evaluate the capability of state-of-the-art LLM-based agents to perform information synthesis and web-based navigation across diverse real-world sources. Accordingly, our design focuses on capturing variation in web content across regions rather than enforcing a uniformly balanced annotator demographic. While annotators were drawn from 11 countries across three continents, the tasks themselves cover **42** countries, and the associated webpages span a broad range of regional domains. As the benchmark evaluates an agent's ability to search, retrieve, and synthesise information from heterogeneous sources, the demographic composition of annotators does not influence the underlying skill being measured.

### A.3 EXAMPLES

In this section (see Table 14), we present some representative examples from DEEPSYNTH bench. We omit some information, e.g. the reasoning trace and intermediate steps, to prevent task leakage.

### A.4 MORE DETAILS ABOUT EVALUATION

**F1 and LLM-Judge Metrics.** F1 in our benchmark is computed using exact string and numeric matching across all fields of the required JSON output. This makes it very strict: even minor deviations (a missing key, a slightly different string form, or a small numerical mismatch) reduce the score sharply. In contrast, the LLM-judge is a soft metric designed to capture partial semantic correctness. It (a) rewards outputs that are semantically equivalent despite surface-level differences (e.g., "U.S." vs. "United States"), and (b) tolerates small numerical deviations (approximately 1%–5.5%), providing a more graded signal of correctness than F1.

Deep Information Synthesis

Figure 7: Annotation Guidelines

**Why EM and LLM Scores?** Exact Match (EM) is well-suited for our benchmark because over 95% of the answers are numeric and all keys correspond to unambiguous factual fields. In this setting, strict matching provides a reliable signal of correctness: either the model produces the correct value for each key or it does not. EM is also deterministic and stable, avoiding the variability or hallucination-related errors that can arise with LLM-based judges.

To complement this strict measure, we additionally use an LLM-based judge that evaluates softer, semantic aspects of the output. This score captures cases where the reasoning is sound and the answers are approximately correct but differ slightly in phrasing or small numerical deviations. Together, EM and the LLM score offer a balanced evaluation: one measures exact factual accuracy, while the other captures approximate correctness and reasoning fidelity.

**Evaluation Example** Below we illustrate how EM, F1, and the LLM score behave under different model outputs.

- **Ground truth:** {"India": 4.5, "China": 7.8, "U.S.": 10.5}

**Model 1 output:**
{"India": 3.6, "China": 8.7, "U.S.": 10.5}
**Scores:** EM = 0.0;  F1 = 33.3;  LLM Score = **1.0**

**Model 2 output:**
{"India": 3.6, "United States": 10.5, "China": 8.7}
**Scores:** EM = 0.0;  F1 = 0.0;  LLM Score = **1.0**

**Model 3 output:**
{"India": 4.5, "U.S.": 10.5, "China": 7.8}
**Scores:** EM = 1.0;  F1 = 100;  LLM Score = **1.0**

## Your Annotation

**Instructions for Providing Answers:**
Use either a JSON object or an Excel sheet. Each entry must include a unique ID and full provenance (data and code).

- Format:
  - Provide answers **either** as a JSON object **or** as an Excel sheet (XLSX or CSV).
- Columns / Keys / Fields Required:
  - **Unique ID**
  - **Question**
  - **Reasoning Path** (include any `code` used)
  - **Answer**
  - **Data file**
- Data Usage:
  - State which data sources were used, download the data locally, and name the downloaded file using the corresponding `Unique ID`.
- Code Inclusion:
  - If code was used, include it verbatim inside the **Reasoning Path** field so the exact commands and scripts are preserved.

**Sample JSON (single item)**

```
{
  "unique_id": "item-001",
  "question": "Compare the gender ratio in two video games and how it changed from 2010 to 2024?",
  "reasoning_path": "1) Search Query: " characters in game X"
                    2) Code used: Calculate the growth rate .. """ def growth(): ... """,
                    3) Notes: used official statistics site (Please DONOT forget to report the exact website)",
  "answer": "X game: 9%, Y game: 15%",
  "data_file": "item-001.csv"
  }
```

**Sample CSV / Excel template (columns)**

| unique_id | question | reasoning_path | answer | url |
|-----------|----------|----------------|--------|-----|
| item-001 | Compare the gender ratio in two video games and how it changed from 2010 to 2024? | see JSON above (includes code) | "X game: 9%, Y game: 15%" | ["https://gameX.com", "https://gameY.com"] |

Figure 8: Annotation Guidelines

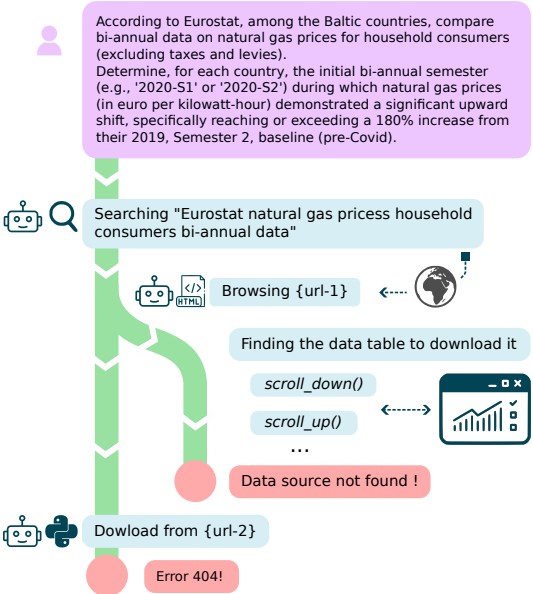

Figure 9: Example run using OWL, illustrating errors when trying to collect and reason about data. The agent finds the right URL but fails to query the right data from the website's database interface. A second attempt tries to download a data file from an incorrect URL, resulting in a *not found error*.

| Questions | Metadata |
|---|---|
| In the 2021 Canadian federal elections, compare the last opinion polling results by Abacus Data, EKOS Research, Mainstreet Research, Forum Research, Research Co. and Nanos Research before the elections. How accurate is each of these polls as a predictor for the actual number of votes for each party (rounded to two decimals), you should include the 6 largest parties, and 'others'. Rank each polling company by mean-squared error. Your answer should be a JSON object with the keys being the polling company names. The values should be the MSE (rounded to one decimal). The companies should be sorted in increasing value of MSE. | `Type`: Compare, Rank
`Region`:North America
`Domain`: Sociopolitics |
| As a Swiss Federal Statistics Officer, I am studying goods transport performance by railway. From 2008 to 2024, what is the average annual change rate for domestic transport, import, export, and transit? Provide the answer as a JSON object where each key is the transport type and each value is the percentage change (rounded to one decimal). | `Type`: Change
`Region`: Europe
`Domain`: Transportation |
| According to ASEAN stats, between 2016-2023, which ASEAN countries' exports in telecommunication, computer and information services had a negative correlation with the total nominal GDP of all ASEAN countries combined? The final answer should be presented as a JSON: keys should be these ASEAN countries, and values should be the pearson correlation value(s) rounded to two decimals. | `Type`: Correlation
`Region`: Southeast Asian
`Domain`: Economics |
| I am analysing the relative importance of airports for Qatar Airways. Using crowd-sourced data from the OpenSky Network for December 2020, can you identify which are the five most important airports for Qatar Airways flights - as measured by their pagerank, using the default damping factor suggested in the paper introducing the google search engine. Your output should be a JSON object with the keys being the IATA airport codes, and the values being the Pagerank value rounded to three decimals. | `Type`: Rank
`Region`: Middle East,
`Domain`: Transportation |
| I am analyzing New Zealand migration data from 2001 to 2020 to identify anomalies in migration patterns. For each year, count the number of months in which the net migration was negative ($< 0$). Please return the results in JSON format, where each key is a year and the value is the number of months with negative net migration. | `Type`: Counting, Compare
`Region`: New Zealand
`Domain`: Socioeconomics |
| What are the tokenizer-level compression ratios (measured as bytes per token) for the following sentence: " Deep Insight Benchmark is an open-source benchmark that evaluates agents' ability to solve tasks requiring analysis of multi-regional and real-world data. ", when tokenized using the tokenizers of Llama (meta-llama/Llama-2-7b-hf), Qwen (Qwen/Qwen3-4B-Base) and Apple (apple/FastVLM-1.5B) models? Return the results as a JSON object, where each key is the model name and the value is the compression ratio (rounded to two decimal places).
{
"Llama":float,
"Qwen":float,
"Apple":float
} | `Type`: Task Specific Quantity
`Region`: None
`Domain`: Computer Science |

Table 14: Examples of questions.

**Model 4 output:**
{"India": 100.7, "U.S.": 100.6, "China": 7.8}
**Scores:** EM = 0.0; F1 = 33.3; LLM Score = 0.0

## A.5 PROMPTS

In this section, we provide the instructions and prompts used across all models; see Figures 10 and 11.

Judge whether the following [response] to [question] is correct or not based on the precise and unambiguous [correct_answer] below.

[question]: {question}
[response]: {response}

Your judgment must be in the format and criteria specified below:

extracted_final_answer: The final exact answer extracted from the [response]. Put the extracted answer as 'None' if there is no exact final answer to extract from the response.

[correct_answer]: {correct_answer}

final answer length: Provide the overall number of unique answers that appear in [response], not just the correct ones. Be sure to provide a number, not an estimate!

reasoning: Explain why the extracted_final_answer is correct or incorrect based on [correct_answer], focusing only on if there are meaningful differences between [correct_answer] and the extracted_final_answer. Do not comment on any background to the problem, do not attempt to solve the problem, do not argue for any answer different than [correct_answer], focus only on whether the answers match.

correct: Answer 'yes' if extracted_final_answer matches the [correct_answer] given above, or is within a small margin of error for numerical problems, a margin of 1 to 5.5 percentage points is acceptable. Answer 'no' otherwise, i.e. if there is any inconsistency, ambiguity, non-equivalency, or if the extracted answer is incorrect.

precision: Answer '1' if extracted_final_answer matches the [correct_answer] given above. Answer '0' otherwise, i.e. if there is any inconsistency, ambiguity, non-equivalency, or if the extracted answer is incorrect. In the case where [correct_answer] is a number or percentage, then answer with the following formula to compute the normalized similarity score:
[1 - (abs([correct_answer] - extracted_final_answer) / max(abs([correct_answer]), abs(extracted_final_answer)))]

final precision: Extract the precision score from above, just the final score (number).

overlapping answers: List all of the answers in [response] that also appear in [correct_answer]. You can consider an answer from [response] to match with an answer in [correct_answer] if it is equivalent or is within a small margin of error for numerical problems, a margin of 1 to 5.5 percentage points is acceptable. List all of the [response] answer appearing in [correct_answer] with each answer delimited by '###'. If the number of overlapping answers is zero, output 'NULL'.

Figure 10: The prompt for the LLM-as-a-judge from Wolfson et al. (2025).

```
INSTRUCTIONS:
- You are a professional researcher preparing a structured,
data-driven answer.
- Your task is to analyze and answer.
- Answers are not report. Often time answers are numeric
and need JSON outputs.
- You will be given a research task by a user. Your task is
to generate an accurate answer.
- Focus on data-rich insights.

GUIDELINES:
1. **Answer Format**
- Be analytical, avoid generalities, and ensure that each
section supports data-backed.
- Prioritize reliable, up-to-date official sources such as
Wikipedia, government websites, etc.
- Every question will already contain the format of the output.
Please follow that.
- Exact match is the metric.
- To help extract the answer, before writing down the final
answer, please using a split token <Answer>:
- It is of utmost importance that you follow the answer format
mentioned in the question.
- If the question says that the model needs to output the
answer JSON. Please follow that.
- If the question says that the model needs to output a list of
JSON. Please follow that.

2. **Incomplete answers**
- Do NOT give incomplete answers.
- There is always an answer, and the answer often needs some
deep reasoning.

3. **Language**
- Please only answer in English
```

Figure 11: System prompt provided to the model, outlining instructions, answer formatting guidelines, and language requirements to ensure structured, data-driven responses.

