# OpenReview forum: "A Benchmark for Deep Information Synthesis"
_ICLR.cc/2026/Conference — ICLR 2026 Poster_

### Official Review · Reviewer_fzgH · 2025-10-29

**Soundness:** 3
**Presentation:** 3
**Contribution:** 3
**Rating:** 6
**Confidence:** 3

**Summary:**

This paper introduces DeepSynth, a very challenging benchmark for evaluating LLM agents. DeepSynth consists of 120 diverse tasks created by 16 experts, where each task requires an agent to navigate through about 4 web pages and read up to 15 documents and tables. The tasks are designed to reflect real-world analysis demand, and cover knowledge across 42 countries. The authors evaluated 5 state-of-the-art models (o4-mini, GPT-4.1, GPT-5, Gemini-2.5-Pro and DeepSeek-R1), as well as 3 state-of-the-art agentic frameworks (o3-deep-research, smolagents and OWL) on DeepSynth. It is observed that this benchmark is challenging for all these models and frameworks, with the best F1 score being only 8.97 points. The authors also conducted a bunch of ablation studies to understand the challenges in DeepSynth. They found models perform worse as the number of intermediate steps increases, and most failures are caused by either navigation error or synthesis error, which suggests future directions for developing LLM agents.

**Strengths:**

1. There lack good benchmarks for developing LLM agents. The DeepSynth benchmark in this paper will help accelerate the research and development of LLM agents.
2. The DeepSynth benchmark is designed to reflect real-world demand. The data curation looks reasonable and solid, though the dataset scale of 120 tasks may be a little bit small.
3. The benchmark is challenging enough even for state-of-the-art models and agentic frameworks. It reveals failures in existing LLM agents and shows substantial room for future improvement.

**Weaknesses:**

1. Information synthesis may not be an intuitive title for the community. To my understanding, what you meant by information synthesis is closely related to multi-step reasoning or agentic tasks. Maybe you can consider agentic tasks as your title?
2. There is a substantial gap between the F1 score and LLM judge score. In my opinion, this indicates that your ground truth answer may have different semantic forms, and string matching doesn’t work well there. Could you evaluate this issue with experiments, and provide some guidelines for those who will use your benchmark?
3. There isn’t a human baseline for this benchmark. A human baseline will let us understand whether the benchmark is solvable for average humans or require specific domain knowledge. I understand this may not be completed during rebuttal since each task requires 5.5 hours for humans on average, but it’s nice to have human baseline results later.

**Questions:**

1. In Figure 5, the space between types should be larger for better readability.

---

> ### Author Response · Authors · 2025-11-20
> **Author Response to Reviewer fzgH**
>
> We thank you for your effort in reviewing our paper and for your thoughtful comments. We are encouraged to see that the reviewer found our  data curation looks reasonable and solid. Please find our detailed response to your concerns outlined below.
>
> > W1: Information synthesis may not be an intuitive title for the community. To my understanding, what you meant by information synthesis is closely related to multi-step reasoning or agentic tasks. Maybe you can consider agentic tasks as your title?
>
> We appreciate the suggestion. While our tasks require multi-step reasoning and the use of external tools, our primary focus is on integrating and synthesising information across multiple sources. This is fundamentally different from prior information-seeking work, as we argue in the Related Work section (lines 469–470). We will also clarify this distinction in the introduction for the camera-ready version.
>
> > W2: There is a substantial gap between the F1 score and LLM judge score. In my opinion, this indicates that your ground truth answer may have different semantic forms, and string matching doesn’t work well there. Could you evaluate this issue with experiments, and provide some guidelines for those who will use your benchmark?
>
> This gap is expected and arises from the fundamental differences between the two metrics.
> F1 performs exact string and number matching across all fields of the required JSON output. This makes it very strict: even minor deviations (a missing key, a slightly different string form, or a small numerical mismatch) reduce the score sharply.
>
> The LLM-judge is a soft metric designed to capture partial semantic correctness. It serves two purposes: (a) it rewards outputs that are semantically equivalent despite minor string differences (e.g., “U.S.” vs “United States” or “Brussels bombing” vs “Brussels bombing attack”), and (b) for numerical answers it allows a small margin of error (we permit differences in the range ~1%–5.5%), producing a more granular and permissible score than EM and F1.
>
> As described in Section Evaluation Setup 3 and illustrated in Figure 6, we explicitly define both the F1 computation and the LLM-judge. We have now added the following example and added a clarification point in the Appendix A.5 (see section Evaluation)
> Below we illustrate how EM, F1, and the LLM score behave under different model outputs.
> ```
> - Original answer: {“India”: 4.5, “China”: 7.8, “U.S”: 10.5}
> - Model 1 output: {“India”: 3.6, “China”: 8.7, “U.S”: 10.5}  EM: 0.0; F1: 33.3; LLM Score: 1.0;
> - Model 2 output: {“India”: 3.6, “United States”: 10.5, “China”: 8.7} EM: 0.0; F1: 0.0; LLM Score: 1.0;
> - Model 3  output: {“India”: 4.5, “U.S”: 10.5, “China”: 7.8}  EM: 1.0 F1: 100; LLM Score: 1.0;
> - Model 4  output: {“India”: 100.7, “U.S”: 100.6, “China”: 7.8}  EM: 0.0 F1: 33.3 ; LLM Score: 0.0;
> ```
> These examples highlight why the F1 metric can under-reward semantically correct responses, while the LLM-judge can correctly differentiate between harmless surface deviations and genuinely incorrect answers. We have added a short clarification in the appendix to guide future users of the benchmark.

---

> > ### Author Response · Authors · 2025-11-20
> >
> > > W3: There isn’t a human baseline for this benchmark. A human baseline will let us understand whether the benchmark is solvable for average humans or require specific domain knowledge. I understand this may not be completed during rebuttal since each task requires 5.5 hours for humans on average, but it’s nice to have human baseline results later.
> >
> >
> > We thank the reviewer for this suggestion. While constructing the benchmark, the annotators' domain knowledge helped us design and verify the tasks. However, we made sure that each question is clearly formulated and contains all the necessary information such that solving them does **not** require prior domain knowledge.
> >
> > Given the limited time during the rebuttal period, we conducted a small human study on 10 randomly sampled tasks. Annotators were provided with the task question and intermediate steps (but not the intermediate or final answers).
> >
> > | Model               | F1-score | LLM_score (GPT-4.1) |
> > |--------------------|----------|-------------------|
> > | Human               | 80       | 80                |
> > | GPT-5               | 10       | 10                |
> > | Gemini-Pro-2.5      | 10       | 10                |
> > | o3-deep-research    | 20       | 20                |
> >
> > The human annotator achieved an F1 score of 80, indicating that the tasks are solvable by humans with guidance from intermediate planning steps. Importantly, completing the tasks does **not** require specific domain knowledge. However, successful completion does require the use of external tools (e.g., web search, browsing, or simple Python code), as expected for models. We plan to expand this human baseline in future work when resources permit.
> >
> >
> >
> > > Q1: In Figure 5, the space between types should be larger for better readability.
> >
> >
> > Thank you for your suggestion. We have updated Figure 5.

---

> > > ### Author Response · Authors · 2025-11-28
> > > **Author Comment**
> > >
> > > Thank you again for reviewing our paper. We hope that our response fully addresses your concerns. If you have any further questions, we would be happy to clarify.

---

### Official Review · Reviewer_hwhP · 2025-11-01

**Soundness:** 4
**Presentation:** 4
**Contribution:** 4
**Rating:** 8
**Confidence:** 3

**Summary:**

This paper proposes DEEPSYNTH, a high-quality, well-designed benchmark comprising 120 challenging and diverse information synthesis tasks across 42 countries and 7 domains. The authors clearly articulate the motivation, methodology, data collection and curation process, followed by a detailed analysis of the dataset characteristics. The paper presents comprehensive experiments evaluating popular SOTA models and specialized agent frameworks, with detailed performance analysis revealing significant limitations in current systems.

Overall, this is a solid paper with rigorous benchmark design and high-quality implementation. The presentation effectively explains the design choices and underlying rationale. The work addresses an important gap in agent evaluation by focusing on information synthesis rather than simple fact retrieval. Recommend accept.

**Strengths:**

- Excellent observation regarding the lack of real-world tasks that require synthesizing information from multiple sources, with strong motivation for designing such a benchmark that is well-suited for advancing current LLM and agent systems which are becoming increasingly powerful and saturating existing benchmarks.
- Dataset collection and curation process is exceptionally well-designed, executed, and presented through a rigorous 4-stage pipeline (data source identification, hypothesis generation, validation, task formulation), enabling readers and users to easily understand the characteristics and quality standards of the benchmark.
- The experiments with SOTA models are comprehensively designed and the analysis and ablations are thoroughly conducted, unveiling the significant incapabilities of current models (best F1 score only 8.97) and underscoring the substantial value and challenge posed by this benchmark.
- The findings derived from the dataset creation process and experimental analysis are insightful and well-supported with quantitative evidence.

**Weaknesses:**

- Overly difficult questions might limit the practical applicability of the benchmark, making evaluations of less powerful models even more challenging. For example, if less powerful models all achieve nearly zero performance, it becomes difficult to obtain meaningful signals to differentiate and evaluate which model performs better. In comparison, benchmarks with graduated difficulty levels generally suffer less from this limitation.
- Since the benchmark requires gathering information from the dynamic Internet that constantly changes, some websites may become unavailable over time (e.g., pages/tables expire or are restructured). This could make certain tasks inaccessible or make long-term maintenance of the benchmark particularly challenging, potentially affecting reproducibility and consistency of evaluations.

**Questions:**

- Regarding the inherently challenging nature of the benchmark, what are the authors' thoughts on improving the applicability so that less powerful models can also derive meaningful evaluation signals from the benchmark? Would incorporating graduated difficulty levels be beneficial?
- Although one design principle emphasizes 'robustness against memorization,' once the benchmark becomes public, won't it inevitably face contamination issues? How do the authors plan to prevent models from over-optimizing specifically for this benchmark, which could cause unfairness when comparing against other models that haven't been exposed to these tasks?

---

> ### Author Response · Authors · 2025-11-20
> **Author Response to Reviewer hwhP**
>
> We thank the reviewer for their highly positive assessment and clear endorsement. We address the questions below.
>
> > W1/Q1: Overly difficult questions might limit the practical applicability of the benchmark, making evaluations of less powerful models even more challenging. For example, if less powerful models all achieve nearly zero performance, it becomes difficult to obtain meaningful signals to differentiate and evaluate which model performs better. In comparison, benchmarks with graduated difficulty levels generally suffer less from this limitation.
>
> We agree that DEEPSYNTH is challenging, and that weaker models may obtain near-zero scores on the full end-to-end tasks. To provide meaningful evaluation signal for such models, we will release a **DEEPSYNTH-Lite/Dev** subset consisting of 40 tasks with the following :
> - **Human-written intermediate steps (planning):** including referenced webpages, intermediate calculations, and code fragments.  [Anonymous Link](https://anonymous.4open.science/r/DEEPSYNTH-FC1C/README.md)
> - **Intermediate answer schemas:** enabling step-level evaluation of planning, retrieval, reasoning, and synthesis. Here is one such example [Anonymous Link](https://anonymous.4open.science/r/DEEPSYNTH-FC1C/decompositions/003.json)
>
> This allows models to be evaluated on **isolated subtasks**, offering more fine-grained diagnostic assessment even when full-task performance is low.
> Importantly, we will not release intermediate steps for the remaining (80) tasks in the main benchmark, as doing so would risk future contamination and reduce the benchmark’s usefulness for long-term agent evaluation.
>
>
>
> > W2: Since the benchmark requires gathering information from the dynamic Internet that constantly changes, some websites may become unavailable over time (e.g., pages/tables expire or are restructured). This could make certain tasks inaccessible or make long-term maintenance of the benchmark particularly challenging, potentially affecting reproducibility and consistency of evaluations.
>
> We agree with the reviewer that relying on the dynamic Internet introduces challenges, as websites and tables may change structure or become temporarily unavailable. To mitigate this, we intentionally selected official and stable data sources (e.g., national statistical agencies, international organizations), which tend to maintain consistent long-term URLs and archival practices.
>
> In addition, we **store snapshots of every referenced table and webpage**. While snapshots cannot fully preserve the original retrieval environment (e.g., live search and navigation behaviour), they *do* ensure that the core reasoning and synthesis steps remain reproducible. This allows future evaluations to maintain consistency even if the live web content evolves.
>
> Together, these choices significantly reduce the impact of web drift and help ensure that DEEPSYNTH remains reproducible and usable over time.
>
>
>
> > Q1: Would incorporating graduated difficulty levels be beneficial?
>
> We agree that having graduated difficulty levels would be beneficial for diagnostic evaluation, especially for weaker models. Defining difficulty automatically/manually is a non-trivial task; difficulty in DEEPSYNTH depends on multiple factors (e.g., number of intermediate steps, number of sources to aggregate, number of tool calls, etc.), not just surface complexity.
>
> As a first step, we will release **DEEPSYNTH-dev** with human-written intermediate steps and metadata such as the **number of intermediate reasoning steps**, which serves as a reasonable proxy for task complexity. This will help practitioners assess model performance across a spectrum of difficulty without altering the integrity of the main benchmark.

---

> > ### Author Response · Authors · 2025-11-20
> >
> > > Q2: Although one design principle emphasizes 'robustness against memorization,' once the benchmark becomes public, won't it inevitably face contamination issues? How do the authors plan to prevent models from over-optimizing specifically for this benchmark, which could cause unfairness when comparing against other models that haven't been exposed to these tasks?
> >
> > We deliberately designed DEEPSYNTH to be resistant to memorization and over-optimization, even after public release. The benchmark inherently requires multi-source reasoning:
> > - Each task depends on up to 15 external sources. This will be challenging to memorise.
> > - Final answers are derived through multi-step calculations, not extractable from any single page.
> > - Outputs require dynamic, multi-field numeric synthesis, rather than natural-language spans that could be memorised.
> >
> > Even if a model memorized the questions, it would still need to reproduce the entire synthesis pipeline using current external data, making direct memorization ineffective.
> > To further mitigate contamination:
> >
> > - We will publicly release 40 tasks (with questions, answers, and intermediate steps) as a supervised DEEPSYNTH-Lite/Dev set. [Anonymous Link](https://anonymous.4open.science/r/DEEPSYNTH-FC1C/DEEPSYNTH_lite.json)
> >
> > - The remaining 80 tasks will be part of a held-out evaluation set that we do not release with answers or steps.
> >
> > - For these held-out tasks, we will release only the questions and maintain a public leaderboard, enabling fair evaluation across models while preventing overfitting.
> >
> >
> > We acknowledge that no benchmark is completely immune to future contamination. However, the design choices above, multi-source derivation, structured numeric outputs, and a held-out evaluation set, substantially mitigate the issue and preserve the fairness and longevity of DEEPSYNTH for evaluating information synthesis capabilities.

---

> > > ### Author Response · Authors · 2025-11-28
> > > **Authors Comment**
> > >
> > > Thank you again for reviewing our paper. We hope that our response fully addresses your concerns. If you have any further questions, we would be happy to clarify.

---

### Official Review · Reviewer_kYyA · 2025-11-01

**Soundness:** 2
**Presentation:** 3
**Contribution:** 2
**Rating:** 2
**Confidence:** 3

**Summary:**

The paper presents DEEPSYNTH, a benchmark of 120 tasks for evaluating LLM agents on multi-source information synthesis across 7 domains and 42 countries. It uses a multi-stage expert-driven pipeline for task creation and shows SOTA agents (e.g., GPT-5, o3-deep-research) achieve low F1 scores (max 8.97), exposing gaps in reasoning and tool use.

**Strengths:**

- Multi-stage pipeline ensures non-memorizable, diverse tasks.
- Covers 9 models/agents with metrics like F1/EM and ablations.
- Breaks down performance by steps/operations.

**Weaknesses:**

- Lacks full code/prompts, data promised post-acceptance.
- Annotator selection biased; uneven regional coverage.
- Overlaps with prior benchmarks without direct comparisons.
- Low baselines may reflect poor prompting.

**Questions:**

- Can you provide inter-annotator agreement metrics (e.g., Cohen's kappa) for hypothesis validation and task formulation to address subjectivity concerns.

---

> ### Author Response · Authors · 2025-11-20
> **Authors Response to Reviewer KYyA**
>
> We thank the reviewer for the constructive comments and for recognising our multi-stage pipeline and breadth of evaluation.
>
> > W1: Lacks full code/prompts, data promised post-acceptance.
>
> We have included the exact prompts used for both the LLM judge and the model evaluations in Figures 6 and 7 and in Appendix A.1. For agent frameworks such as OWL [1] and Smolagent [2], we strictly followed their documented configurations.
> In line with the venue’s policy, **we will release the full benchmark (dev/test) and evaluation scripts immediately upon acceptance**. We have also added illustrative examples to the (currently private) repository to help reviewers understand the pipeline: [Anonymous DEEPSYNTH bench](https://anonymous.4open.science/r/DEEPSYNTH-FC1C/README.md)
>
> [1] https://github.com/camel-ai/owl/tree/main
> [2] https://github.com/huggingface/smolagents
>
> > W2: Annotator selection biased; uneven regional coverage.
>
> Thank you for raising this point. The purpose of our benchmark is to evaluate the ability of state-of-the-art LLM-based agents to perform **information synthesis and web-based navigation** across diverse real-world sources. Our focus was therefore on capturing variation in *web content* across regions, rather than on achieving perfectly balanced annotator demographics.
> It is correct that annotators came from 11 countries across three continents, and this distribution is not uniform. However, the benchmark tasks themselves span **42 countries**, and the underlying webpages are drawn from a broad set of regional domains. The demographic distribution of annotators does not affect the core capability we aim to evaluate: **whether an agent can search, retrieve, and synthesise information from heterogeneous sources**, regardless of the region of origin. We agree that this is a valuable point. We have updated and clarified this in the paper (See Appendix A.3 Lines 798).
>
> > W3: Overlaps with prior benchmarks without direct comparisons.
>
> In Table 6, we provide a comparison and emphasise that: GAIA, AssistantBench, BrowseComp, and HLE focus on factual retrieval or goal-directed search, whereas DEEPSYNTH focuses on multi-source numerical synthesis, derivation, and structured outputs. These benchmarks do not contain comparable task formats (multi-document → numeric synthesis → JSON), so a direct comparison is not applicable.
>
> The comparison we presented may not fully match the type of analysis the reviewer expected. We would be grateful if the reviewer could specify the form of comparison they believe would be most helpful. This would allow us to address it appropriately.
>
> > W4: Low baselines may reflect poor prompting.
>
> All models were evaluated using carefully optimised prompts, and all agentic frameworks were run with their official recommended settings. We have included the exact prompts used for both the LLM judge and the model evaluations in Figures 6 and 7 and in Appendix A.1. For agent frameworks such as OWL [1] and Smolagent [2], we strictly followed their documented configurations.
> To ensure fairness, we manually inspected 10 randomly sampled failure cases per model and found **no evidence of prompt-induced systematic bias**. As noted in Lines 308–319, current agent systems often fail to use external tools reliably, and DEEPSYNTH tasks cannot be solved using parametric knowledge alone, highlighting the need for stronger tool-using agents rather than prompt tuning.

---

> ### Author Response · Authors · 2025-11-20
>
> > Q1: Can you provide inter-annotator agreement metrics (e.g., Cohen's kappa) for hypothesis validation and task formulation to address subjectivity concerns?
>
> Thank you for raising the concern about subjectivity. During data construction, hypothesis formulation, and task formulation, identifying relevant data sources and deriving the required intermediate quantities is a time-consuming and challenging process. For this reason, each hypothesis was generated by a single expert annotator, rather than collecting multiple independent hypotheses for the same source. A second expert then performed independent validation and cross-checking of the hypothesis, verifying every intermediate step and the final derived answer. The validator also flagged any subjective or ambiguous formulations, and such cases were revised or removed.
>
> Because the pipeline uses a creator–validator structure rather than two fully independent creators, there are no two parallel annotations for the same hypothesis. As a result, standard inter-annotator agreement metrics (e.g., Cohen’s κ) do not apply, and we therefore did not compute an inter-annotator agreement score for hypothesis validation.
>
>
> Given the limited time during the rebuttal period, we conducted a small human study on 10 randomly sampled tasks. Annotators were provided with the task question and intermediate steps (but not the intermediate or final answers).
>
> | Model               | F1-score | LLM_score (GPT-4.1) |
> |--------------------|----------|-------------------|
> | Human               | 80       | 80                |
> | GPT-5               | 10       | 10                |
> | Gemini-Pro-2.5      | 10       | 10                |
> | o3-deep-research    | 20       | 20                |
>
> The human annotator achieved an F1 score of 80, providing evidence that the tasks are clear, unambiguous, and solvable when presented with the same high-level planning structure that models receive. This supports our claim that subjectivity in task formulation has been effectively controlled by the creator–validator pipeline.

---

> > ### Author Response · Authors · 2025-11-28
> > **Authors Comment**
> >
> > Thank you again for reviewing our paper. We hope that our response fully addresses your concerns. If you have any further questions, we would be happy to clarify.

---

### Official Review · Reviewer_Pai7 · 2025-11-01

**Soundness:** 3
**Presentation:** 3
**Contribution:** 3
**Rating:** 6
**Confidence:** 4

**Summary:**

This paper introduces DeepSynth, a benchmark that evaluates agent on realistic and time consuming problems. DeepSynth covers 120 tasks across 7 domains and data sources covering 42 countries.  Each task requires agents to navigate ~4.2 web pages, and read between 1 to 15 context documents. The benchmark has been proven to be challenging even by state-of-the-art models like GPT-5 and Gemini-Pro-2.5 and agentic systems like o3-deep-research and smolagent, showing DeepSynth can serve as an arena of evaluating the capabilities of agentic systems.

**Strengths:**

- The construction process of DeepSynth is rigorous. It consists of five steps: (1) data source identification, (2) hypothesis generation, (3) hypothesis validation, (4) task formulation, and (5) data validation. All steps is done by expert annotators. The demographic information of the annotators & average time consumption are also reported in this paper.
- DeepSynth is challenging for many state-of-the-art models and systems, showing huge gap between human capabilities and current models.
- The analysis section gives interesting insights, and also provides insights on how the agents fail, whether the groudtruth intermediate steps will help, and how the agent behaves across tasks from different regions.

**Weaknesses:**

- I understand the benchmark is challenging. However, the evaluation metrics are hard to interpret (especially EM and LLM Judge Score, see the questions section). The value of EM is almost all 0s and LLM Judge Score might favor their own model family (GPT-4.1).
- (minor) The annotator demographic might also be biased (75% male, 81.25% PhD).

**Questions:**

- Is EM reliable as a metric? The correlation between EM score and LLM Judge Score are not strong. Is there some examples where the EM is low but the LLM Judge Score is high & EM high but LLM Judge score low?
- Is the model used for LLM Judge Score GPT-4.1 (the same in Wolfson et al. (2025))? In this case, is it likely that it is biased towards its own generation? How do we interpret that Smolagent (GPT-4.1) has higher LLM Judge Score than Smolagent (GPT-5), but F1 scores show otherwise.
- Thank you for providing the time estimation! What are the approximate cost of compute for running different agents on this benchmark?
- Is there an inter-annotator agreement rate for hypothesis validation?
- Is the human performance 100 among experts since DeepSynth only include tasks where answers from both annotators were identical?
- With only 120 tasks, what are the confidence intervals on the
reported scores?

---

> ### Author Response · Authors · 2025-11-20
> **Authors Response to Reviewer Pai7**
>
> We thank the reviewer for the thoughtful and constructive feedback. We address all concerns below.
>
> > **W1:**  I understand the benchmark is challenging. However, the evaluation metrics are hard to interpret (especially EM and LLM Judge Score, see the questions section). The value of EM is almost all 0s, and LLM Judge Score might favour their own model family (GPT-4.1).
> > **Q1:** Is EM reliable as a metric? The correlation between EM score and LLM Judge Score are not strong. Is there some examples where the EM is low but the LLM Judge Score is high & EM high but LLM Judge score low?
>
> **Q1.1: Is EM reliable as a metric?**
>
> We appreciate the reviewer’s concern. All tasks in our benchmark require models to output in  **JSON format** with string keys and values that are strings, numbers, or lists of numbers. More than 95% of answers are numeric, and all keys are unique factual descriptors. As defined in our Task Construction Criteria (Sec. 2.1), every question in DEEPSYNTH is designed to have a unique, fully verifiable correct answer.
>
> In this setting, exact matching is appropriate: a model must return the correct structured fields and the correct numeric values. EM provides a clear signal of strict correctness.  Unlike LLM-based judges) EM is deterministic, reproducible, and unaffected by hallucination or bias.
>
> The LLM-judge, in contrast, captures partial semantic correctness and rewards outputs that are close but not exact. The two metrics, therefore, serve complementary purposes: EM measures precise factual correctness, while the LLM-judge measures how much of the structure and content the model gets right when it falls short of exactness. Together, they provide a more complete view of model performance.
>
>
> **Q1.2 Why can EM and LLM-judge differ substantially?** A gap occurs when a model gets most of the structured output correct but makes a few minor errors: EM becomes 0 (since EM requires every field to match exactly), while the LLM-judge still assigns a high score because the response is largely correct semantically.
> Below is an illustrative example:
> ```
> Original answer:
> {“India”: 4.5, “China”: 7.8, “U.S”: 10.5}
> Model 1 Output:
> {“India”: 3.6, “China”: 8.7, “U.S”: 10.5}
> EM: 0.0 | F1: 33.3 | LLM-judge: 1.0
> Model 2 Output:
> {“India”: 3.6, “United States”: 10.5, “China”: 8.7}
> EM: 0.0 | F1: 0.0 | LLM-judge: 1.0
> Model 3 Output:
> {“India”: 4.5, “U.S”: 10.5, “China”: 7.8}
> EM: 1.0 | F1: 100 | LLM-judge: 1.0
> Model 4 Output:
> {“India”: 100.7, “U.S”: 100.6, “China”: 7.8}
> EM: 0.0 | F1: 33.3 | LLM-judge: 0.0
> ```
> Please note that LLM-judge is a soft metric designed to capture partial semantic correctness. It serves two purposes: (a) it rewards outputs that are semantically equivalent despite minor string differences (e.g., “U.S.” vs “United States” or “Brussels bombing” vs “Brussels bombing attack”), and (b) for numerical answers it allows a small margin of error (we permit differences in the range ~1%–5.5%), producing a more granular and permissible score than EM and F1.
>
> **This pattern explains observations in Table 2 as well.** In Table 2, o3-deep-research has a modest EM (2.5) but a much higher LLM-judge (17.5), whereas Smolagent (GPT-4.1) shows EM = 2.5 and LLM-judge = 7.5. This difference reflects the type of errors made: o3-deep-research typically commits many small, localised mistakes (e.g., one field rounded slightly differently or one key missing), so its outputs are mostly correct and the LLM-judge awards substantial partial credit. Smolagent, by contrast, makes larger or more systematic errors across multiple fields, which lowers its judge score even when EM is the same.
>
> We will add it to the result section, a camera-ready version of the paper.
>
> **> W2. Annotator demographic bias (minor)**
>
> Thank you for raising this point. The purpose of our benchmark is to evaluate the ability of state-of-the-art LLM-based agents to perform **information synthesis and web-based navigation** across diverse real-world sources. Our focus was therefore on capturing variation in *web content* across regions, rather than on achieving perfectly balanced annotator demographics.
>
> It is correct that annotators came from 11 countries across three continents, and this distribution is not uniform. However, the benchmark tasks themselves span 42 countries, and the underlying webpages are drawn from a broad set of regional domains. The demographic distribution of annotators does not affect the core capability we aim to evaluate: whether an agent can search, retrieve, and synthesise information from heterogeneous sources, regardless of the region of origin. We agree that this is a valuable point. We have updated and clarified this in the paper (See Appendix A.3 Lines 798).

---

> > ### Author Response · Authors · 2025-11-20
> >
> > >Q2.1:**Is the model used for LLM Judge Score GPT-4.1 (the same in Wolfson et al. (2025))? In this case, is it likely that it is biased towards its own generation?**
> >
> > |              Model        |   F1-score |   EM      | LLM_score (GPT4.1) |LLM_score (GPT-5) |LLM_score (Gemini-Pro-2.5) |
> > |---------------------------|---------------|------------|-----------------------------|---------------------------|---------------------------------------|
> > | Smolagent (GPT-4) |     3.75     |     2.50  |            7.5                  |          7.5                 |                    5.0                     |
> > | Smolagent (GPT-5) |     6.42     |    1.67   |            2.5                  |          2.5                 |                    3.3                     |
> > | Gemini-Pro-2.5       |      6.25     |     0.0    |            5.0                 |           5.0                 |                    5.0                     |
> >
> > We follow Wolfson et al. (2025) exactly to maintain comparability. To explicitly assess potential bias, we re-evaluated all tasks using GPT-4.1, GPT-5, and Gemini-2.5-Pro as the judge. The differences in scores across judges were minor, and there was no consistent trend favouring GPT-4.1 outputs.  Thus, while minor variations exist due to differences in scoring style, the results indicate that using GPT-4.1 as the judge does not systematically bias the evaluation toward outputs from GPT-4.1 models.
> >
> > >Q2.2 **How do we interpret that Smolagent (GPT-4.1) has higher LLM Judge Score than Smolagent (GPT-5), but F1 scores show otherwise.**
> >
> > This observation is closely related to our explanation in **Q1.2**. In general, F1 is sensitive to surface-form deviations (e.g., different phrasing, ordering of keys), while the LLM-judge rewards semantic and structural correctness even when the output is not an exact match.
> > However, numerical tasks introduce an additional nuance:  the LLM-judge allows a small tolerance window for numeric values (~1%–5.5%). If the model’s numeric errors fall outside this window, the LLM-judge assigns a score of 0, even if some fields are correct. In contrast, F1 still gives partial credit for the correctly matched key–value pairs.
> > This example illustrates the point:
> > ```
> > - Original: {“India”: 4.5, “China”: 7.8, “U.S”: 10.5}
> > - Model output: {“India”: 100.7, “U.S”: 100.6, “China”: 7.8}
> > - Scores: EM = 0.0 | F1 = 33.3 | LLM-judge = 0.0
> >
> > ```
> > Here, one country (“China”) is correct, so F1 gives partial credit. But because the other numeric errors are significant, the LLM-judge assigns a score of 0.
> >
> > This difference explains why a model may have a higher LLM-judge score while having a lower F1 score (or vice versa), depending on whether the model makes small, localised deviations or large numerical errors.
> > We have updated the paper with the details in Appendix A.3.
> >
> > >Q3. Approximate compute cost (What are the approximate cost of compute for running different agents on this benchmark?)
> >
> > We provide an approximate total cost for running 120 tasks (based on 2025 API pricing):
> >
> > |**Model** |**Cost**| **Output Token (Min and Max)**
> > |----------|----------|----------|
> > | deepseek-R1 | $0.041  |687.4 (avg)|
> > | GPT-4 | $0.432  | 188 (min)  383 (max) |
> > | GPT-5 | $5.52 | 1600(min) 15872(max) reasoning tokens, 201(min) - 425 (max) completion tokens|
> > | o4-mini| $1.39 | 448 (min) - 5760 (max) reasoning tokens, 40 (min) - 392 (max) completion tokens|
> > | o3 deep-research| $184.61 | 448 (min) - 5760 (max) reasoning tokens, 40 (min) - 392 (max) completion tokens|
> > |Gemini-Pro-2.5|$11.78|12995 (avg)|
> >
> > We have added the compute-cost detail in Table 8 (see Appendix Page 12).

---

> ### Author Response · Authors · 2025-11-20
>
> >Q4. Inter-annotator agreement during hypothesis validation (Is there an inter-annotator agreement rate for hypothesis validation?)
>
> During data construction, hypothesis formulation involved identifying relevant data sources and deriving the required intermediate quantities, which is a time-consuming and challenging process. For this reason, each hypothesis was generated by a single expert annotator, rather than collecting multiple independent hypotheses for the same source. A second expert then performed independent validation and cross-checking of the hypothesis, verifying every intermediate step and the final derived answer.
>
> Because the pipeline uses a creator–validator structure rather than two fully independent creators, there are no two parallel annotations for the same hypothesis. As a result, standard inter-annotator agreement metrics (e.g., Cohen’s κ) do not apply, and we therefore did not compute an inter-annotator agreement score for hypothesis validation.
>
> >Q5. Human performance (Is the human performance 100 among experts since DeepSynth only include tasks where answers from both annotators were identical?)
>
> During data validation, a second annotator independently answered each drafted question. For this step, the annotator was given the 130 drafted tasks along with the underlying data sources and intermediate steps (but not the final answers). We retained 120 tasks for which both annotators produced identical final answers. This agreement check ensures that the ground-truth answers are reproducible and unambiguous. However, it is not intended to serve as a human performance estimate on the full benchmark: annotators in this phase were not required to perform navigation, search, or synthesis from scratch.
>
> A complete human evaluation of the full pipeline: starting only from the task question and performing all retrieval, analysis, and synthesis, is both expensive and time-intensive. As noted in the paper, creating each task required ~5.5 hours on average. Running a full human baseline under these conditions is infeasible within the review cycle.
>
> Given the limited time available, we conducted a small human study on 10 randomly sampled tasks. Annotators were given the task question and the intermediate planning steps (but not intermediate or final answers), and asked to complete the tasks.
>
> | Model              | F1-score | LLM-score (GPT-4.1) |
> |--------------------|----------|----------------------|
> | **Human**          | **80**   | **80**               |
> | GPT-5              | 10       | 10                   |
> | Gemini-Pro-2.5     | 10       | 10                   |
> | o3-deep-research   | 20       | 20                   |
>
> The human annotator achieved an F1 score of 80, indicating that the tasks are solvable when humans are provided with the same high-level planning structure that models receive.
>
> Importantly, this small study is not intended as a substitute for a full human baseline. A fair human performance estimate would require humans to start without any provided planning steps and would likely benefit from a clearly defined time limit to ensure comparability. Our goal in this study was simply to verify that all questions are **solvable by humans**, not to establish a definitive human score on the benchmark.
>
>
> >W6. With only 120 tasks, what are the confidence intervals on the reported scores?
>
> We evaluate all models using pass@1 to reflect realistic usage, where agents provide a single answer per task. To assess the stability of the results, we repeated the experiments three times and computed the variance.
>
> | Model | F1-score | LLM_score (GPT4.1)|
> |----------|----------|----------|
> | GPT-4 | 3.6 +/- 0.5 | 0.0 |
> | GPT-5 | 3.9 +/- 0.3 |  0.0 |
> | Gemini-Pro-2.5 | 5.8 +/- 0.5 | 5.0 +/- 2.5|
> | Smolagent (GPT-5) | 6.7 +/- 0.7 |  5.0 +/- 2.5 |
>
> The reported intervals indicate that variance across runs is generally low, suggesting that model performance is stable. Notably, the LLM-judge scores show higher variance for models with partial semantic correctness, reflecting the metric’s sensitivity to subtle differences in reasoning outputs.
>
> We thank the reviewer again for the helpful comments.

---

> > ### Author Response · Authors · 2025-11-28
> > **Authors Comment**
> >
> > Thank you again for reviewing our paper. We hope that our response fully addresses your concerns. If you have any further questions, we would be happy to clarify.

---

### Author Response · Authors · 2025-12-03
**Summary for AC**

We thank all reviewers once again for their constructive feedback.

The reviewers found the DeepSynth benchmark to be challenging (Pai7, fzgH), important (hwhP), and timely (fzgH). They also recognised our dataset construction methodology as rigorous and comprehensive (Pai7, fzgH, kYyA).

To address the concerns raised by the reviewers, we have added new experimental results covering:

- Approximate computational cost of running different agents on the benchmark (Pai7).
- Potential bias toward model-generated data and whether the benchmark favours its own generation (Pai7).
- Variance of LLM performance on the benchmark (Pai7).
- Human performance, including when humans are provided with the reasoning chains (Pai7, fzgH, kYyA).

In addition, we expanded and clarified the description of our evaluation metrics to resolve the following points:

- How to interpret cases where Smolagent (GPT-4.1) attains a higher LLM-Judge score but a lower F1 score than Smolagent (GPT-5) (Pai7, fzgH).
- Clarification that “Information Synthesis” is thematically consistent with the goals of our paper (fzgH).

We also note that we have fully addressed the concerns raised by reviewer kYyA. We now provide an anonymous link to the dev set of the benchmark. The prompts used for model evaluation and generation were already included in the submission.

In the revised version, we have incorporated all the requested results and clarifications. These additions are reflected in the Appendix, with all new text highlighted in blue, and we have included Table 8 to report the computational cost.

We believe DeepSynth will be a valuable resource for the community, supporting the development and evaluation of agentic frameworks and proprietary models.

---

### Meta-Review · Area_Chair_wh54 · 2025-12-23

**Summary:**

This paper introduces DeepSynth, a benchmark of 120 tasks evaluating LLM agents on multi-source information synthesis across 7 domains and 42 countries. The benchmark demonstrates that state-of-the-art models like GPT-5 and o3-deep-research achieve modest F1 scores of at most 8.97, revealing substantial gaps in current agent capabilities. Reviewers recognized several strengths: the rigorous five-stage construction pipeline with expert annotators, the challenging nature of tasks requiring navigation across multiple webpages and synthesis from up to 15 documents, comprehensive evaluation of 9 models with detailed ablation studies, and the benchmark's focus on numerical derivation rather than simple fact retrieval. Weaknesses identified include potentially difficult-to-interpret evaluation metrics, particularly the gap between EM and LLM-judge scores, limited scale with only 120 tasks raising questions about statistical confidence, concerns about reproducibility given reliance on dynamic web content, and initial lack of code, prompts, and human baseline performance. The most important reasons for acceptance are that DeepSynth addresses a timely gap in agent evaluation by requiring genuine multi-source synthesis rather than memorizable retrieval, the construction methodology is sound and well-documented, and the benchmark reveals meaningful limitations in current systems that warrant community attention.

**Reviewer Concerns:**

The authors provided substantive responses addressing nearly all reviewer concerns. They added computational cost analysis showing o3-deep-research costs approximately $185 for 120 tasks while more efficient models like DeepSeek-R1 cost only $0.04. To address potential bias, they re-evaluated tasks using GPT-4.1, GPT-5, and Gemini-2.5-Pro as judges, finding no systematic bias favoring GPT-4.1 outputs. They reported variance across three runs showing stable performance with low variance in F1 scores. Most significantly, they conducted a small human study on 10 tasks where humans achieved 80% F1 when provided intermediate planning steps, compared to 10-20% for best models, confirming tasks are solvable and well-formulated. They also released an anonymous dev set link with examples and clarified metric interpretation through concrete examples showing how EM, F1, and LLM-judge capture different aspects of correctness. Reviewer Pai7's concerns about metric interpretation and computational costs were thoroughly addressed with detailed examples and cost breakdowns. Reviewer hwhP's questions about difficulty levels and contamination were addressed by committing to release a 40-task dev set with intermediate steps while holding out 80 tasks for fair evaluation. Reviewer fzgH's concerns about the gap between F1 and LLM-judge scores were clarified with illustrative examples, and the human baseline concern was addressed with the small study. Reviewer kYyA's concerns about missing code, prompts, and inter-annotator agreement were addressed by providing the dev set link, exact prompts in appendices, and explaining why standard agreement metrics don't apply to their creator-validator pipeline. The human study providing 80% F1 effectively demonstrates task clarity despite lack of traditional agreement metrics. Outstanding concerns are minor: kYyA's desire for more direct benchmark comparisons remains somewhat unresolved as authors argue different task formats make direct comparison inappropriate, and the small scale of 120 tasks limits granular analysis though variance studies show stability. In weighing these points, the substantive additions (human baseline, cost analysis, bias testing, dev set release) combined with clear metric clarifications significantly strengthen the submission and address the core reproducibility and interpretability concerns.

**Reviewer Scores:**

Pai7 initially scored 6 and would likely maintain or increase to 6 or possibly 8 given their concerns were comprehensively addressed with new experimental results, detailed metric explanations, and cost analysis that directly answered their questions. Reviewer hwhP scored 8 and explicitly recommended acceptance, so would maintain 8 as their concerns were constructively addressed with the dev set commitment and contamination mitigation strategy. Reviewer fzgH scored 6 and would likely maintain 6 or increase to 8 since the metric interpretation was clarified with concrete examples and the human baseline was provided despite time constraints. Reviewer kYyA scored 2 but would likely move to 4 or 6 given the substantial response addressing reproducibility through dev set release, prompt transparency, and human baseline demonstration, though the lack of follow-up engagement suggests uncertainty about full satisfaction.

---

### Decision · Program_Chairs · 2026-01-26

Accept (Poster)